# Aerosol layer height (ALH) retrievals from oxygen absorption bands: intercomparison and validation among different satellite platforms, GEMS, EPIC, and TROPOMI

**Hyerim Kim**[1,2], **Xi Chen**[1,2], **Jun Wang**[1,2,3,4], **Zhendong Lu**[2,3,4], **Meng Zhou**[5], **Gregory R. Carmichael**[1,2], **Sang Seo Park**[6], **and Jhoon Kim**[7]

[1]Department of Chemical and Biochemical Engineering, The University of Iowa, Iowa City, IA 52242, USA
[2]Center for Global and Regional Environmental Research (CGRER), The University of Iowa, Iowa City, IA 52242, USA
[3]Iowa Technology Institute, The University of Iowa, Iowa City, IA 52242, USA TS1
[4]Interdisciplinary Graduate Program in Informatics, The University of Iowa, Iowa City, IA 52242, USA
[5]Goddard Earth Sciences Technology and Research (GESTAR) II, University of Maryland – Baltimore County, Baltimore, MD 21228, USA
[6]School of Urban and Environmental Engineering, Ulsan National Institute of Science and Technology (UNIST), Ulsan 689-798, Republic of Korea
[7]Department of Atmospheric Sciences, Yonsei University, Seoul 03722, South Korea

**Correspondence:** Xi Chen (xi-chen-4@uiowa.edu) and Jun Wang (jun-wang-1@uiowa.edu)

**Abstract.** The vertical distribution of aerosols is crucial for assessing surface air quality and its impact on the climate. Although aerosol vertical structures can be complex, assuming a certain shape for the aerosol vertical profile allows for the retrieval of a single parameter – aerosol layer height (ALH) – from passive remote sensing measurements. In this study, we evaluate ALH products retrieved using oxygen absorption measurements from multiple satellite platforms: the Geostationary Environment Monitoring Spectrometer (GEMS) focusing on Asia, the Earth Polychromatic Imaging Camera (EPIC) in deep space, and the polar-orbiting TROPOspheric Monitoring Instrument (TROPOMI). We use the extinction-weighted aerosol optical centroid height (AOCH) derived from aerosol extinction profiles of Cloud-Aerosol Lidar with Orthogonal Polarization (CALIOP) as the ground truth. The differences due to the inconsistent definitions of ALH in various retrieval algorithms are investigated and eliminated before comparison. We select multiple dust and smoke cases under ideal observational conditions, referred to as "golden days", for the evaluation. Given the significant role of aerosol optical depth (AOD) in ALH retrieval, we first evaluate the AOD from these retrievals against the ground-based AErosol RObotic NETwork (AERONET). Results show that the GEMS AOD at 440 nm has better agreement with the AERONET AOD of the $\sim 0.9$ correlation coefficient ($R$) than that at 680 nm, both of which underestimate with a negative bias. In contrast, EPIC and TROPOMI tend to overestimate AOD by 0.33 and 0.23 for dust cases, while the bias for smoke plumes is small. Evaluation of ALH against CALIOP demonstrates that the EPIC/TROPOMI ALH has good consistency ($R > 0.7$) with CALIOP but is overestimated by approximately 0.8 km. The GEMS ALH displays minimal bias (0.1 km) but a slightly lower correlation ($R = 0.64$). Intercomparisons between three passive retrievals indicate that GEMS retrievals have a limited consistency with EPIC and TROPOMI of 0.3–0.4 $R$, while GEMS underestimates with ALHs of $\sim 0.3$ and $\sim 0.6$ km compared with TROPOMI and EPIC, respectively. The correlations improve under conditions of higher absorbing aerosols (UVAI $\geq 3$), as the signal in the oxygen absorption band ($O_2$–$O_2$ used by GEMS) is enhanced. Although the ALH diurnal cycle from EPIC and GEMS shows some differences, they both demonstrate ALH descent in the afternoon, which might be related to the boundary layer process. Case stud-

ies show that the EPIC ALH indicates a morning ascent to around 4.5 km, while the GEMS ALH remains stable before descending to below 3 km in the afternoon.

# 1   Introduction

Atmospheric aerosols influence Earth's energy budget and climate system by absorbing and scattering solar and terrestrial radiation (Wang and Christopher, 2003). The aerosol vertical distribution is one of the most important factors determining aerosol radiative effects (Zhang et al., 2013). The altitudes of absorbing aerosols such as dust and smoke affect the vertical distribution of radiative heating and modify the stability of the atmosphere (Babu et al., 2011; Koch and Del Genio, 2010; Wendisch et al., 2008; Wang and Christopher, 2006). When aerosols are lifted to high altitudes in the upper troposphere and lower stratosphere, they can have longer residence times and transports over longer distances, influencing the global radiative energy budget (Christian et al., 2019; Peterson et al., 2014). The aerosol vertical distribution influences the derivation of aerosol optical properties, such as aerosol optical depth (AOD) and single scattering albedo (SSA) in the ultraviolet (UV) spectrum, where the top-of-atmosphere (TOA) radiance is also sensitive to the vertical variation of aerosols (Torres et al., 1998). Furthermore, estimates of the surface concentration of particulate matter (PM) from the total columnar aerosol loading or AOD require knowledge or assumptions about the aerosol vertical distribution (Wang and Christopher, 2003). The aerosol profile is controlled by diverse processes, such as convective transport, in-cloud scavenging, particle growth by condensation, biomass burning emission and injection height, and boundary layer mixing, depending on the different sources and aerosol properties (Wang et al., 2006; Kipling et al., 2013; Yang et al., 2013; Kipling et al., 2016). Due to the complexity of these processes and the lack of temporally and spatially resolved information, the aerosol vertical profile has high uncertainty and diversity in chemical transport models (Wang et al., 2013; Yang et al., 2013; Koffi et al., 2016). Therefore, measuring an accurate aerosol vertical distribution is still a challenge but is critical in many research areas.

Satellite remote sensing techniques are effective for globally monitoring aerosol vertical profiles. Active satellite remote sensing, especially by a spaceborne lidar such as Cloud-Aerosol Lidar with Orthogonal Polarization (CALIOP) on board the Cloud-Aerosol Lidar and Infrared Pathfinder Satellite Observation (CALIPSO) platform, acquires backscatter profiles and retrieves aerosol extinction profiles with high vertical resolution (Winker et al., 2013). However, the global coverage of CALIOP is less than 0.2 % due to its narrow swath and wide gaps between orbits. In contrast, the larger spatial coverage of passive remote sensing measurements overcomes this shortcoming. With the retirement of

CALIPSO in August 2023, passive remote sensing has become the only routine technique accessible to the public at present from space for filling the data gap when measuring aerosol vertical distributions before the next lidars dedicated to measuring aerosols are launched into space. However, only limited information on aerosol extinction vertical profiles can be obtained through passive remote sensing due to the need for multiple assumptions regarding the surface and aerosol properties of the retrieval process (Geddes and Bösch, 2015; Rao et al., 2022). Several parameters, including the spectral coverage, radiance, polarization, spectral resolution, signal-to-noise ratio (SNR), and number of viewing angles, can influence the information content and retrieval uncertainties of aerosol profiles (Chen et al., 2021a).

Hence, many algorithms have been developed to extract a single piece of information regarding aerosol vertical distribution, with a primary emphasis on aerosol layer height, which approximates the altitude of aerosols of a presumed aerosol vertical profile. Passive sensing techniques to retrieve aerosol layer height (ALH) information include stereo photogrammetry, polarimetric techniques in the UV–VIS spectrum, the infrared (IR) technique, and atmospheric oxygen ($O_2$) absorption (Pierangelo et al., 2004; Muller et al., 2007; Zeng et al., 2008; Vandenbussche et al., 2013; Wu et al., 2016; Xu et al., 2018; Kim et al., 2023). Not only are these techniques based on different physical theories, but each product has different definitions of ALH and parameterizations of aerosol profiles, including aerosol optical central height (AOCH) and aerosol effective height (AEH). Part of this study will analyze how the assumption of the shapes of aerosol vertical profiles in the retrieval may lead to inherent differences in the retrieval product. Beyond this theoretical analysis, this study mainly focuses on evaluating three ALH data products retrieved from three different satellite sensors that detect the TOA measurements in various $O_2$ absorptions from the visible to near-infrared bands.

Aerosols positioned at lower altitudes cause light to travel a longer path, resulting in increased absorption by $O_2$ molecules along the extended path (Ding et al., 2016; Xu et al., 2019). Consequently, the amount of scattered radiation received by satellites decreases as the aerosol layer decreases in altitude. Kokhanovsky and Rozanov (2010) retrieved the top height of the dust layer by fitting spectral TOA reflectance measurements from the $O_2$ A band (around 760 nm) of the Scanning Imaging Absorption spectroMeter for Atmospheric CHartographY (SCIAMACHY). Similarly, the official operational ALH product of the TROPOspheric Monitoring Instrument (TROPOMI) uses measurements of the $O_2$ A band to retrieve the centroid pressure or height of a presumed single aerosol layer (Nanda et al., 2020). However, retrieving ALH information over land, including areas with vegetation and soil surfaces, from the $O_2$ A band presents challenges. This is because the TOA reflectance in this band is dominated by high surface reflectance instead of aerosol scattering. Consequently, TOA reflectance becomes less sen-

sitive to ALH, and errors in surface reflectance contribute to significant uncertainties in ALH retrieval (Xu et al., 2019).

Despite the weaker oxygen absorption in the $O_2$ B band (near 688 nm) than in the $O_2$ A band, the surface reflectance is significantly lower in the $O_2$ B band across all the land types, which proves advantageous for aerosol retrieval. Based on this principle, Xu et al. (2017, 2019) developed a retrieval algorithm that uses measurements from both the $O_2$ A and B bands, applying it to observations from the Earth Polychromatic Imaging Camera (EPIC) and Deep Space Climate Observatory (DSCOVR) to produce a product known as aerosol optical central height (AOCH). Based on this algorithm, with several adjustments, an enhanced algorithm was developed and implemented in hyperspectral measurements from TROPOMI (Chen et al., 2021b) as TROPOMI AOCH. In comparison with the operational TROPOMI ALH product, which shows a bias of 2 km over land, the TROPOMI AOCH shows a bias of approximately 0.5 km over both ocean and land (Chen et al., 2021b; Nanda et al., 2020). Hence, in the context of this study, we employ the TROPOMI AOCH retrieved through the combined utilization of the $O_2$ A and B bands, favoring it over the operational TROPOMI ALH product retrieved solely from the $O_2$ A band.

Furthermore, oxygen-dimer ($O_2$–$O_2$) absorption bands exhibit a sensitivity to ALH that is similar to $O_2$ absorption bands. Aimed at observing in the 300 to 500 nm range, the Geostationary Environment Monitoring Spectrometer (GEMS) can measure radiation across multiple $O_2$–$O_2$ absorption bands, including 340, 360, 380, and 477 nm, with 477 nm found to be the most sensitive to the ALH due to its highest $O_2$–$O_2$ absorption (Chimot et al., 2017; Cho et al., 2024; Kim et al., 2020; Park et al., 2016). GEMS provides an aerosol layer height product, termed aerosol effective height (AEH), retrieved from the 477 nm $O_2$–$O_2$ absorption band. This algorithm has been applied in Ozone Monitoring Instrument (OMI) measurements and was recently evaluated with CALIOP, revealing negligible bias and a standard deviation of 1.4 km in the AEH difference across the GEMS observation domain from January to June 2021 (Park et al., 2023, 2016).

The three oxygen-related bands, i.e., the $O_2$–$O_2$ 477 nm band and the $O_2$ A and B bands, have differences in terms of oxygen absorption strength and surface reflectance, leading to their diverse sensitivities to ALH. Hence, comparing retrievals from different $O_2$ absorption bands can offer valuable insights into their respective advantages and limitations in ALH retrieval. This motivates us to validate three different satellite ALH products, i.e., GEMS, EPIC, and TROPOMI, using CALIOP's three-dimensional aerosol extinction product, and to conduct intercomparisons between them. While the validation of diurnal variations in ALH currently remains challenging, leveraging GEMS' hourly products alongside the near-hourly EPIC global retrievals allows us to conduct comparative analyses with the available data at hand.

Additionally, evaluation of ALH retrievals should consider the context of other retrieval parameters, such as AOD and the UV aerosol index (UVAI). The UVAI quantifies the difference between measured and calculated near-UV spectral dependence, with values near zero indicating an aerosol-free atmosphere or the presence of non-absorbing aerosols and clouds, while positive values are associated with UV-absorbing aerosols like carbonaceous aerosols, volcanic ash, and desert dust (Torres et al., 2007). Accurate retrieval of ALH requires reliable retrieval of AOD since the retrieval sensitivity is highly dependent. Park et al. (2016) showed higher sensitivity of oxygen-dimer ($O_4$) slant column density to aerosol effective height at 477 nm with a higher AOD. Xu et al. (2019) showed that the sensitivity of differential optical absorption spectroscopy (DOAS) ratios – the ratio of TOA reflectance between the absorption band and the continuum band – to ALH is enhanced for lower surface reflectance and higher AOD values. Moreover, the sensitivity of the UVAI to ALH, along with their correlation, rises with increasing AOD levels (Xu et al., 2019, 2017). Therefore, we evaluate AOD with the AErosol RObotic NETwork (AERONET) and cross-compare the AOD and UVAI products from different platforms together with the relationship of ALH between products with different UVAI values to see their impact.

Considering that the spatial coverage of CALIOP is limited, we carefully selected "golden" cases where dust and smoke events favor retrievals from all three sensors. This selection can maximize the SNR for ALH retrieval, and hence the evaluation can shed light on future improvement to bring about closure of various types of retrievals. Note that these conditions may differ from those observed on non-selected days. In addition to pixel-by-pixel comparison of these passive satellite products, they are also assessed with CALIOP aerosol extinction profiles along CALIOP's track. We provide a detailed comparison with CALIOP profiles for a dust and smoke plume case. This paper outlines the data and comparison approach in Sect. 2, followed by the comparison results for all the data used in this study in Sect. 3. Section 4 shows the investigation of the ALH variation during transport for the selected dust and smoke cases. Lastly, the conclusions and discussions are provided in Sect. 5.

## 2 Data and methodology

The ALH products compared in this study share similarities in that they are all derived using oxygen (or its dimer) absorption bands and assume the same aerosol vertical profile shape. However, there are distinct variations in the specifics of each algorithm, including the definition of ALH, which may result in inherent differences in ALH retrievals. In Sect. 2.1, we first introduce the characteristics of each passive product, providing some details of each retrieval algorithm and presenting the retrieval performance from previous studies. The difference in ALH definitions is compared

in Sect. 2.2. Lastly, the approaches for comparing ALH data and evaluating them with ground-based observations or active measurements are shown in Sect. 2.3.

## 2.1 Remote sensing data

### 2.1.1 GEMS/GK-2B

From a geostationary orbit about 36 000 km above the Equator, GEMS provides hourly measurements over Asia within the latitudes of 5° S to 45° N and the longitudes of 80 to 152° E (Kim et al., 2020). Given the lower SNR in the morning due to a large solar zenith angle (SZA), GEMS only scans the eastern half of the field of view, leading to fewer products being available over the western region. The total number of hourly products on each day also depends on the SZA in different seasons. The spatial resolution of the GEMS products is $3.5 \times 8$ km (north–south and east–west) at Seoul, South Korea.

GEMS offers two products describing aerosol altitude, AEH, and aerosol loading height, each derived from different algorithms. The GEMS aerosol loading height, included in the level-2 GEMS aerosol product (L2AERAOD), employs an optimal estimation method incorporating measurements at six wavelengths, including the $O_2$–$O_2$ band at 477 nm (Cho et al., 2024; Kim et al., 2018). In contrast, the GEMS AEH algorithm uses the sensitivity of the $O_2$–$O_2$ band to the ALH similarly to TROPOMI and EPIC with the $O_2$ A and B bands, which will be discussed in the following subsection. Therefore, this study specifically focuses on analyzing GEMS AEH version 2.0.

The GEMS AEH is retrieved using the $O_2$–$O_2$ slant column density (SCD) at 477 nm with a lookup-table (LUT) approach adopting aerosol types, AOD, and SSA at 550 nm from L2AERAOD and surface reflectance from the GEMS standard product for surface reflectance (Park et al., 2023). Three aerosol types were classified using the UVAI and the visible aerosol index (VisAI), which, similar to the UVAI but with visible channels, categorize aerosols as highly absorbing fine (HAF), dust, and non-absorbing (NA). NA aerosols are selected when the UVAI yields a negative value, the dust type is determined when both the UVAI and VisAI are positive, and the HAF type is selected when the UVAI is positive but the VisAI is negative (Cho et al., 2024). For LUT generation, aerosols are assumed to be spherical due to the computationally intensive spectral-binning method, and the particle size distribution, refractive index, and fine-mode fraction for each aerosol type are derived from the global AERONET inversion climatology (Cho et al., 2024).

Cho et al. (2024) validated the GEMS AOD at 443 nm against AERONET data across the entire GEMS domain from 1 November 2021 to 31 October 2022. They found that the total GEMS AOD showed an $R$ value of 0.792, a root mean square error (RMSE) of 0.227, and a mean bias error (MBE) of 0.038. Park et al. (2023) retrieved and validated

the GEMS AEH with the CALIOP AEH. The differences in AEH between GEMS and CALIOP for the dust plume cases were $-0.07 \pm 1.09$ and $-0.11 \pm 1.27$ km, with 53.8 % and 72.9 % of all the pixels showing differences of less than 1.0 and 1.5 km, respectively. Moreover, during the period from January to June 2021, they observed an average AEH difference of $-0.03$ km (Park et al., 2023).

### 2.1.2 EPIC/DSCOVR

Carried on the DSCOVR spacecraft at the Sun–Earth Lagrangian-1 (L1) point, $1.5 \times 10^6$ km from Earth, EPIC captures images for the sunlit disk of Earth every 60–100 min d$^{-1}$. As a result, EPIC monitors the half-globe near-hourly, rendering a full disk of $2048 \times 2048$ pixels at a spatial resolution of 12 km at Earth's surface (Marshak et al., 2018). With 10 narrow channels, EPIC detects the Earth-reflected solar radiance from the ultraviolet, visible, and near-infrared (NIR) bands, including both the $O_2$ A and B bands. The lower surface reflectance in the $O_2$ B band compared to the $O_2$ B band over land suggests that the $O_2$ B band can be used to improve the ALH retrievals with the $O_2$ A band only (Xu et al., 2019).

Xu et al. (2017) developed an algorithm to retrieve aerosol optical central height (AOCH) from EPIC measurements in the $O_2$ A and B bands for the first time and applied it to dust plumes in the Atlantic Ocean. Later, Xu et al. (2019) added a smoke model to the LUT and applied it to several smoke plume cases over the Hudson Bay–Great Lakes area in North America. They found that over 77 % of collocated AOD pairs fell within an uncertainty envelope of $\pm(0.05 + 0.1 \text{ AOD})$, with a coefficient of determination ($R^2$) of 0.54 (Xu et al., 2019). Based on this algorithm, Lu et al. (2021) updated the calibration of EPIC level-1 data and analyzed the EPIC AOCH for US smoke plumes during the 2020 California large wildfires. The validation of the EPIC AOCH against the extinction-weighted AOCH from lidar observations (CALIOP) in these papers shows a high level of accuracy, with a correlation coefficient of 0.885 and a RMSE of 0.92 km for absorbing aerosols. The surface reflectance data involve two sources: land surface reflectance is obtained from the MODIS surface bidirectional reflectance climatology, while water surface reflectance is derived from the GOME-2 surface Lambertian-equivalent reflectivity (LER) database. Furthermore, a new LUT developed specifically for dust plumes in the East Asian region, based on multi-year AERONET inversion products, has been incorporated, as detailed in Lu et al. (2023).

### 2.1.3 TROPOMI

TROPOMI on board the Copernicus Sentinel-5 Precursor (S5P) satellite was launched in October 2017 to measure solar radiation reflected by Earth from the UV to short-wave infrared (SWIR) bands. This spectral range includes

many trace gas absorption bands and is also sensitive to aerosols and surface properties. Flying on a polar satellite, TROPOMI provides global atmospheric component products at a high spatial resolution of $5.5\,km \times 3.5\,km$ (improved from $7\,km \times 3.5\,km$ since August 2019) once every day.

TROPOMI measures both the $O_2$ A and B absorption bands, yet its official ALH product only utilizes the $O_2$ A band measurements in its retrieval algorithm (Nanda et al., 2020). Chen et al. (2021b) developed an alternative algorithm suitable for TROPOMI data, enabling AOCH retrieval using both the $O_2$ A and B bands. This approach draws upon the EPIC AOCH retrieval algorithm of Xu et al. (2019), employing the same LUT and least-square method to optimize the AOCH from the ratio of $O_2$ absorption to its nearby continuum band. Enhancements include spectral-resolution convolution into multiple narrow channels, a new cloud mask, and dust/smoke classification, with results reported in a standard latitude–longitude grid ($0.05° \times 0.05°$). Comparative analysis reveals that the AOCH exhibits a bias of approximately 0.5 km over both ocean and land, contrasting with the 2 km bias observed in the operational ALH product from TROPOMI (Chen et al., 2021b; Nanda et al., 2020). Consequently, this study employs TROPOMI AOCH retrieval data, as previously highlighted in the Introduction section. Furthermore, the new LUT developed for Asian dust plumes in EPIC retrievals has been integrated into the algorithm for application in East Asia. The surface reflectance data are the same used in the EPIC retrieval algorithm. The operational TROPOMI level-2 UVAI (340–380 nm) product (Stein Zweers, 2022) is used to retrieve only pixels covered by absorbing aerosols with a UVAI greater than 0.5.

### 2.1.4 CALIOP/CALIPSO

CALIOP is a lidar system on the CALIPSO platform that provides attenuated backscatter vertical profiles of aerosols and clouds in the atmosphere using a two-wavelength laser operating at 532 nm with linear polarization and at 1064 nm (Winker et al., 2009). While the global coverage of CALIOP is less than 0.2 %, it provides high vertical resolution for retrieving aerosol extinction profiles (Winker et al., 2013). In this paper, we used the CALIOP 5 km level-2 aerosol extinction profile product at 532 nm to derive optical-depth-weighted heights. Specifically, the Level 2 Aerosol Profile, Version 4-21 data product for the year 2021 (https://doi.org/10.5067/CALIOP/CALIPSO/CAL_LID_L2_05kmAPro-Standard-V4-21, NASA/LARC/SD/ASDC, 2018b) was used. For the years 2022 to 2023, the Level 2 Aerosol Profile, Version 4-51 (https://doi.org/10.5067/CALIOP/CALIPSO/CAL_LID_L2_05kmAPro-Standard-V4-51, NASA/LARC/SD/ASDC, 2025) was used due to the data availability. To validate aerosol height retrievals from passive remote sensing with CALIOP observation, the optical-depth-weighted heights derived from the CALIOP 5 km level-2 aerosol extinction

profile product at 532 nm following previous studies were used (Lu et al., 2023; Chen et al., 2021b; Lu et al., 2021; Xu et al., 2019).

### 2.1.5 AERONET

AERONET is a ground-based remote sensing network designed to measure and characterize aerosol optical properties through direct Sun measurements taken with Sun- and sky-scanning spectral radiometers (Holben et al., 1998). AERONET serves as a critical tool for validating satellite-retrieved aerosol optical properties, including AOD. In this study, we used AOD data at 675 and 440 nm from AERONET Version 3 Level 1.5 to assess the accuracy of satellite AOD retrievals. AERONET sites located within our study domain of East Asia and Southeast Asia, as illustrated in Fig. 2, were selected for this analysis. Additional information on these sites can be found in Table S1 in the Supplement.

### 2.2 Comparison of ALH definitions

The GEMS, TROPOMI, and EPIC algorithms all operate under the assumption of a quasi-Gaussian vertical distribution of aerosol extinction described by the parameters loading, peak height ($H$), and half-width ($\eta$) fixed at 1 km. The assumption of a 1 km half-width is grounded in typical lidar observations for dust and smoke aerosols, as indicated by Reid et al. (2003). This value has also been used in the retrieval of AOD from UV observations by both the Total Ozone Mapping Spectrometer (TOMS) and OMI, as emphasized in the work by Torres et al. (1998). Presently, it is widely accepted as a standard parameter value, as evident in the products presented in this study. The aerosol extinction profile where $z$ is the altitude with respect to the surface can be expressed by a generalized distribution function as specified in Eq. (1):

$$\beta(z) = W \frac{\exp(-\sigma_H |z - H|)}{\left[1 + \exp(-\sigma_H |z - H|)\right]^2}, \tag{1}$$

where $H$ is the altitude with peak aerosol extinction, $W$ is the normalization constant related to the columnar loading, and $\sigma_H$ is the half-width parameter defined as $\sigma_H = \ln\left(3 + \sqrt{8}\right)/\eta$ (Spurr and Christi, 2014). However, ALH is a general term for describing the altitude of an aerosol layer, while the definition of a retrieved ALH varies by algorithm. EPIC and TROPOMI defined their retrieved ALH as $H$ in Eq. (1) and referred to it as AOCH (Chen et al., 2021b; Lu et al., 2023, 2021; Xu et al., 2019, 2017). The GEMS AEH is defined as the altitude above which the aerosol extinction is the $1/e$ of the total columnar AOD, as described in Eq. (2):

$$\frac{\int_0^{AEH} \beta(z)\,dz}{\int_0^{TOA} \beta(z)\,dz} = 1 - e^{-1}. \tag{2}$$

In addition, the AOCH retrieved from EPIC and TROPOMI O2AB-UI is relative to the geographical (ground) surface,

whereas the GEMS AEH is relative to sea level. Henceforth, for simplicity and consistency, the term ALH will be used to refer to all the aerosol height products used in this study, including the GEMS AEH, TROPOMI AOCH, and EPIC AOCH. To validate the retrievals from passive remote sensing with lidar data, the optical-depth-weighted heights derived from CALIOP are used. We define the CALIOP AOCH as the optical-depth-weighted height as specified in Eq. (3):

$$\text{AOCH}_{\text{CALIOP}} = \frac{\sum_{i=1}^{n} \beta(z_i) \Delta z_i z_i}{\sum_{i=1}^{n} \beta(z_i) \Delta z_i}, \tag{3}$$

where $\beta(z_i)$ represents the 532 nm aerosol extinction coefficient at vertical level $i$ with an altitude of $z$, while $\Delta z_i$ denotes the thickness of the vertical layer $i$.

The comparison of different definitions of ALH for the same aerosol vertical distribution is shown in Fig. 1. In an example aerosol extinction profile with the EPIC/TROPOMI AOCH at 1.5 km, the CALIOP AOCH is higher at 1.65 km, with the GEMS AEH being highest at 1.87 km (Fig. 1a). The difference between the EPIC/TROPOMI and CALIOP AOCHs decreases as the AOCH increases, ultimately disappearing when the CALIOP AOCH reaches approximately 4 km and above. When the CALIOP AOCH is below $\sim 1$ km, the EPIC/TROPOMI AOCH can be as much as 0.5 km TS2 lower than CALIOP. The GEMS AEH exhibits a larger difference compared to the CALIOP AOCH for a higher AOCH, and this difference remains relatively constant at approximately 0.3 km for altitudes above $\sim 3$ km. Figure 1c illustrates that the difference between the GEMS AEH and the EPIC/TROPOMI AOCH can reach around 0.8 km near the surface. However, this difference decreases as the AOCH increases, ultimately reaching 0.1 km for altitudes above $\sim 3$ km. In our further comparison of ALH, we account for these inherent differences by converting one definition into another to ensure consistency. Varying the AOCH from 0 to 10 km, we created a LUT of the AEH, EPIC/TROPOMI AOCH, and CALIOP AOCH that corresponded to the same aerosol extinction profile according to their different definitions. Throughout this paper, we carried out two conversions to ensure consistency: first, we converted all the passive-sensor ALH products using the CALIOP AOCH definition for comparison with the CALIOP data (Fig. 1b). Second, we converted the GEMS AEH according to the EPIC/TROPOMI AOCH definition for comparisons between passive remote sensing products (Fig. 1c).

## 2.3 Comparison approach

Given the availability of EPIC/TROPOMI retrievals for absorbing aerosols, we focus our comparison on a selection of golden days characterized by ideal viewing conditions for the dust and smoke cases, excluding cloud-covered areas, as observed within the GEMS field of view from 2021 to 2023. These selected days are listed in Table 1 and correspond to the study domain depicted in Fig. 2. Classification of the dust

**Table 1.** Case study dates and the number of observations from each sensor for each case.

| Case no. | Date (yyyy-mm-dd) | Domain | Number of orbits (or granules) | | |
|---|---|---|---|---|---|
| | | | GEMS | TROPOMI | EPIC |
| D1* | 2021-03-28 | East Asia | 7 | 2 | 4 |
| D2 | 2021-04-26 | East Asia | 8 | 2 | 7 |
| D3 | 2022-04-10 | East Asia | 8 | 2 | 5 |
| D4 | 2023-03-10 | East Asia | 7 | 2 | 3 |
| D5 | 2023-05-19 | East Asia | 8 | 2 | 8 |
| D6 | 2023-05-20 | East Asia | 8 | 2 | 7 |
| B1 | 2021-03-31 | Southeast Asia | 6 | 1 | 5 |
| B2 | 2021-08-10 | East Asia | 8 | 1 | 8 |
| B3 | 2021-08-11 | East Asia | 6 | 2 | 8 |
| B4 | 2022-04-09 | Southeast Asia | 8 | 2 | 5 |
| B5 | 2023-03-26 | Southeast Asia | 6 | 1 | 5 |
| B6 | 2023-04-17 | Southeast Asia | 8 | 1 | 4 |

* The initial "D" represents the dust case, and "B" represents the smoke case from biomass burning.

and smoke cases is determined by the predominant aerosol type identified in the CALIOP lidar data. Although the EPIC products have a similar temporal resolution to GEMS, observations over the research domain vary from 3 to 8 d$^{-1}$, depending on the solar geometry. Flying in a polar orbit, TROPOMI only observes the whole globe once each day, but depending on the latitude of each case, the TROPOMI ground track may overlap, which can lead to the possibility of two TROPOMI observations for some cases. Considering the differences in the spatial and temporal resolutions between these three products, we first resample the GEMS product to the EPIC/TROPOMI spatial resolution using the pixel-area-weighted method and then linearly interpolate the GEMS product to match the observation time for the paired EPIC/TROPOMI data (Wang et al., 2020).

UVAI data from GEMS, EPIC, and TROPOMI are also cross-compared as the UVAI is used in all the retrieval algorithms. The ALH comparison and validation for different GEMS UVAI values are conducted to analyze the possible distinction of GEMS AEH retrieval accuracy for different UVAI values. Furthermore, as the accuracy of each AOD product also influences the corresponding ALH retrieval, AOD will be validated using the ground-based AERONET inversions. When matching satellite pixels with ground sites, we consider the number of valid satellite retrievals within a 0.2° radius around AERONET sites. If the number of valid retrievals exceeds 30 % of the total number of pixels, we compute the mean value of these retrievals and compare it with the corresponding AERONET AOD. The AERONET AOD is averaged for a period of 30 min before and after each satellite observation, aligning with the satellite overpass time (or observation time for GEMS). Furthermore, we only include satellite data points with a spatial standard deviation of less than 0.3 to ensure spatial consistency in the comparison. Since the EPIC/TROPOMI AODs are retrieved at 680 nm,

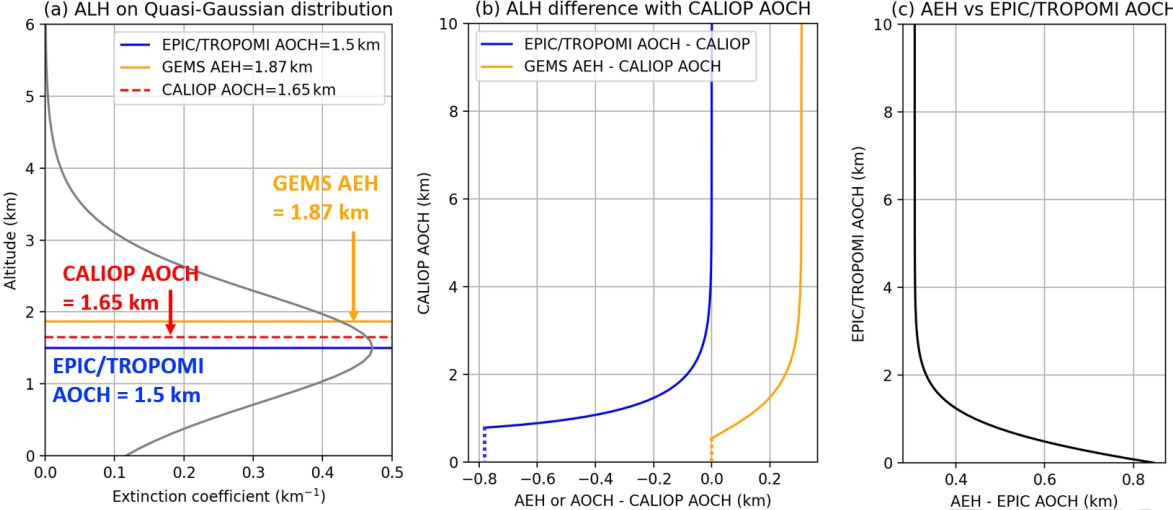

**Figure 1.** Comparison of ALH definitions (GEMS, EPIC, TROPOMI, and CALIOP). **(a)** Relative heights within the quasi-Gaussian distribution when the EPIC/TROPOMI AOCH is 1.5 km. **(b)** Difference between the ALH from the passive satellite and the CALIOP AOCH. Note that the EPIC/TROPOMI AOCH is depicted as dotted vertical lines when it becomes negative below a specific CALIOP AOCH. **(c)** Difference between the GEMS AEH and EPIC/TROPOMI AOCH definitions with respect to the altitude of the EPIC/TROPOMI AOCH.

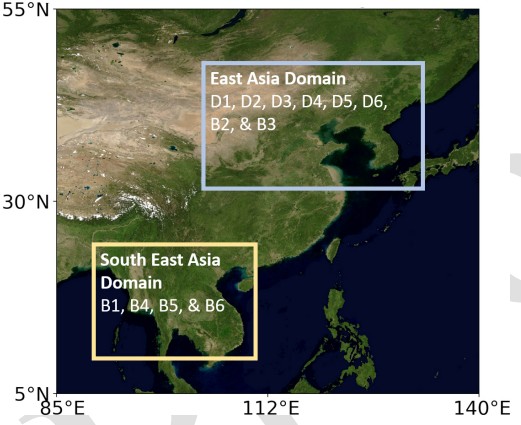

**Figure 2.** Study domains and cases. "D" represents dust cases, while "B" represents smoke cases ("B" stands for biomass burning). See Table 1 for more information. The map is from Blue Marble: Next Generation from the NASA Earth Observatory.

whereas the GEMS AOD is retrieved at 443 nm, we estimate the GEMS AOD at 680 nm from its AOD at 443 nm. This estimation is based on a combination of the aerosol type (dust, HAF, or NA) for each pixel and the Ångström exponent (440–677 nm) from the GEMS aerosol model corresponding to the aerosol type (Kim et al., 2018). When collocating passive satellite products with CALIOP pixels along the track, we employ a similar approach to the comparison with AERONET. This involves calculating the distance from the center of the CALIPSO ground track within a range of 0.2° and adjusting the threshold for valid retrieval to exceed 30 %.

## 3 Results

For all the dust and smoke cases listed in Table 1, the AOD products from GEMS, TROPOMI, and EPIC are first validated against the ground-based AERONET AOD data. Subsequently, a pixel-by-pixel intercomparison is conducted among the satellite products. Additionally, the ALH products from the three passive satellite measurements undergo validation using the CALIOP level-2 aerosol extinction profile. These validated ALH products are then intercompared.

### 3.1 AOD intercomparison and validation with AERONET

The validation of the AOD products from GEMS, TROPOMI, and EPIC against the AERONET AOD is shown in Fig. 3. The GEMS AOD at 443 nm exhibits a strong positive correlation with the AERONET AOD at 440 nm, with correlation coefficients ($R$s) of 0.9 for the dust cases and 0.88 for the smoke cases (Fig. 3a). At 680 nm, the correlation for the smoke cases remains high at $R = 0.84$, indicating a similar level of agreement with the 443 nm measurements. However, for the dust cases at 680 nm, the correlation decreases to $R = 0.73$, along with a 17 % increase in the RMSE, indicating distinct retrieval accuracy of the GEMS AOD at 443 and 680 nm for dust. The GEMS AOD at 680 nm shows higher underestimation than at 443 nm, which is particularly noticeable when the AERONET AOD exceeds 0.5. Furthermore, the bias increases with higher AOD levels, as shown in Fig. S1. The observed underestimation of the GEMS AOD at 680 nm can be in part due to an overestimation of the Ångström exponent (AE), which can be affected by an in-

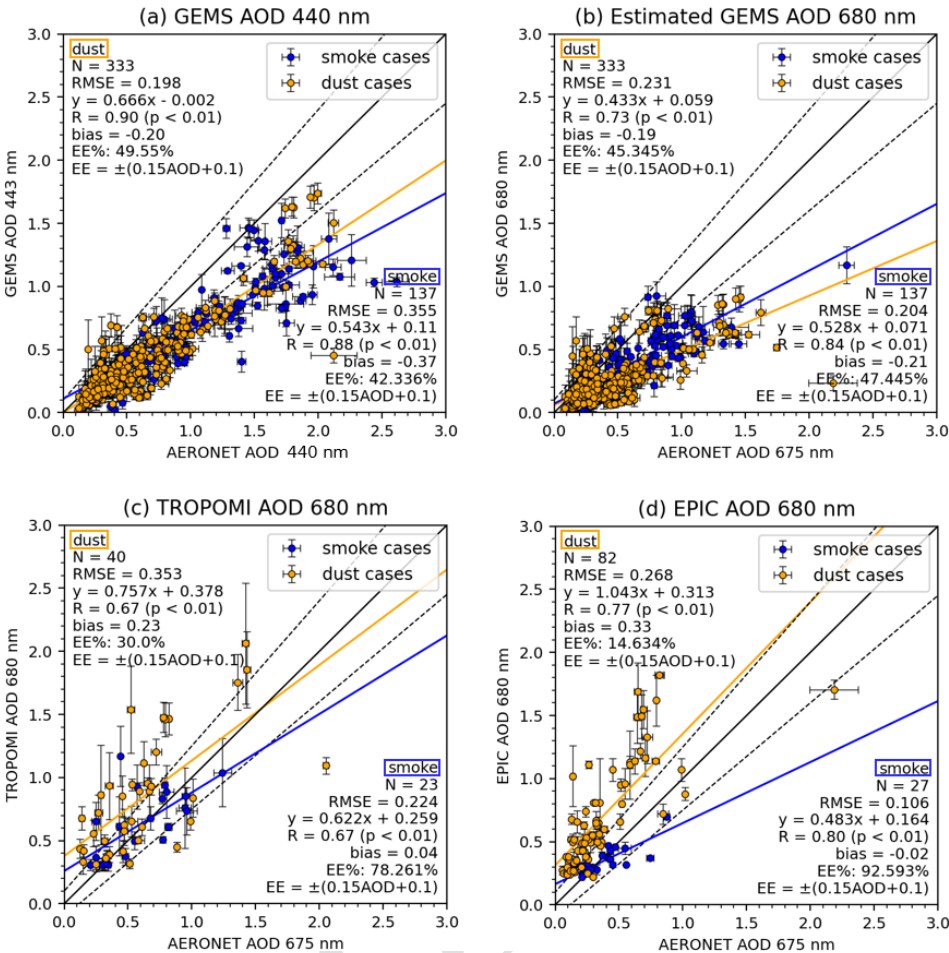

**Figure 3.** Comparison of the GEMS, TROPOMI, and EPIC AODs with the AERONET AOD for all the cases. The blue and yellow dots denote the smoke and dust cases, respectively. **(a)** Comparison of the GEMS AOD at 443 nm with the AERONET AOD at 440 nm. Comparison of the **(b)** estimated GEMS AOD at 680 nm, **(c)** TROPOMI AOD at 680 nm, and **(d)** EPIC AOD at 680 nm with the AERONET AOD at 675 nm. The solid black line is the 1 : 1 line, the colored solid lines are the regression lines, and the dotted lines are the error envelopes for AOD: EE = ±(0.15 AOD + 0.1). Annotated are the number of scatter points (*N*), root mean square error (RMSE), correlation coefficient (*R*), significance level (*p*), mean bias, and percentage of data points within the error envelope (EE). Satellite data points with a standard deviation of less than 0.3 are shown for spatial consistency.

accurate particle size or refractive index in the wavelength-dependent aerosol model.

For the dust cases, both the TROPOMI and EPIC AODs show a positive bias compared to the AERONET AOD, with values of 0.23 and 0.33 for TROPOMI and EPIC, respectively. In contrast to the dust cases, the TROPOMI and EPIC AODs exhibit a negligible bias and a smaller RMSE for the smoke cases. Although the TROPOMI and EPIC AODs do not provide retrievals for values less than 0.2, many AERONET AOD data points exist with values below this threshold, particularly in the dust cases. This suggests that the surface reflectance employed in the dust aerosol model from TROPOMI and EPIC may be underestimated, resulting in an overestimation in the AOD retrieval. For TROPOMI and EPIC retrievals over land, climatological surface reflectance data from MODIS are employed. Additionally, un-

like the GEMS AEH retrieval algorithm that uses GEMS level-2 surface reflectance data, Cho et al. (2024) developed a new method for GEMS AERAOD product retrieval, employing a novel hourly surface reflectance database generated through the minimum reflectance method, which integrates climatological minimum reflectance values for each pixel within a ±15 d window over a 2-year period along with monthly background AOD data. This novel surface reflectance estimation from GEMS AOD retrieval is shown to be effective. In summary, GEMS consistently underestimates AOD, especially at 680 nm, compared to the AERONET AOD. EPIC and TROPOMI, while tending to overestimate AOD in the dust cases due to the underestimated surface reflectance, show a more accurate dust aerosol model than for smoke.

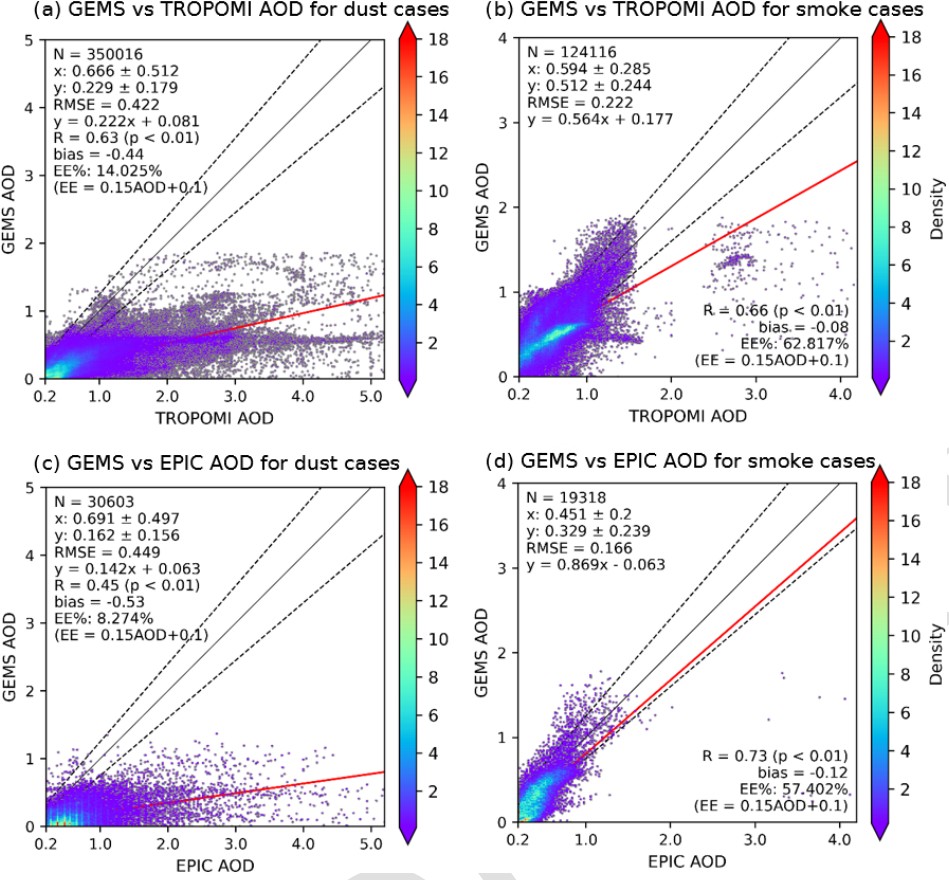

**Figure 4.** The GEMS AOD compared with the corresponding TROPOMI and EPIC products for the dust and smoke cases. **(a)** Scatter density plot of the GEMS AOD versus the TROPOMI AOD for the dust cases. **(b)** Same as **(a)** but for the smoke cases. **(c)** The GEMS AOD versus the EPIC AOD for the dust cases. **(d)** Same as **(c)** but for the smoke cases. The grey points indicate points where the data density is less than 0.01. The black solid line is the 1 : 1 line, and the red solid line is the regression line. The dotted lines indicate error envelopes ($EE = \pm 0.15$ $AOD + 0.1$). The TROPOMI and EPIC AODs do not have retrievals for values less than 0.2; therefore, the figure axes start from where the data exist.

Following the validation with the AERONET AOD, Fig. 4 shows a comparison of the GEMS AOD and the TROPOMI and EPIC AODs, presented separately for the dust and smoke cases. For the dust cases, the GEMS AOD is significantly lower compared with TROPOMI and EPIC, with negative mean biases of $-0.44$ and $-0.53$ for EPIC and TROPOMI, respectively. The inaccuracy of the GEMS dust aerosol model, as identified in the previous AERONET validation (Fig. 3), has a notable impact on the significant difference of the GEMS AOD with TROPOMI and EPIC. Furthermore, surface reflectance issues observed in TROPOMI and EPIC contribute to this disparity. Specifically, surface reflectance estimates for land surfaces from TROPOMI and EPIC may need refinement, as GEMS AOD values close to zero tend to be higher in TROPOMI and EPIC over land but are not observed over water (Fig. S3). The smoke cases show higher agreement compared to the dust cases, as indicated by decreased negative biases from $-0.44$ ($-0.53$) to $-0.08$ ($-0.12$) and RMSE values from 0.42 (0.45) to 0.22

(0.17) for TROPOMI (EPIC). The agreement is particularly robust between GEMS and EPIC, as indicated by a high $R$ value of 0.73. While the smoke aerosol model employed in TROPOMI and EPIC is not as effective as the dust model, its impact on the comparison is relatively minor. These factors in the aerosol model, including aerosol properties, the fine-mode fraction, the phase function, and the SSA, can greatly influence the accuracy of AOD retrievals. Furthermore, AOD sensitivity to changing SSA and surface reflectance is shown in Fig. S4. A detailed description of the aerosol models employed in the AOD retrieval is provided in Table S2.

## 3.2 UVAI intercomparison

The UVAI products of the three satellites are compared, since the UVAI is used as a criterion to focus on retrieving absorbing aerosols for the current TROPOMI and EPIC AOCH. Figure 5 compares the GEMS UVAI with the TROPOMI UVAI (Fig. 5a) and the EPIC UVAI (Fig. 5b) for all the

cases. Compared with TROPOMI, the GEMS UVAI is systematically higher with a positive mean difference of 1.21. GEMS compared to EPIC also exhibits a positive bias, although to a lesser extent (0.32), and shows a higher correlation ($R = 0.57$). This suggests that, when employing the UVAI as a filtering criterion for identifying absorbing aerosols in the EPIC and TROPOMI AOCH retrievals, it is important to establish a threshold that considers the differences between these distinct products. The differences in the GEMS, TROPOMI, and EPIC UVAI products can have several causes. First, different wavelengths are used to derive the UVAI product: the GEMS UVAI is derived from radiances at 354 and 388 nm, whereas the TROPOMI and EPIC UVAIs are derived from radiances at 340 and 380 nm. Additionally, pressure assumption and resolution differences can cause the differences in their UVAI retrievals. We include the EPIC and TROPOMI UVAI comparison scatterplot in Fig. S5. To summarize, the GEMS UVAI is systematically higher than TROPOMI and is more comparable to EPIC.

## 3.3  ALH validation with CALIOP

To ensure comparison of the same variable, it is critical to account for differences arising from the different ALH definitions as detailed in Sect. 2.2. As such, the ALH values of all the passive sensors are converted into AOCH following the CALIOP AOCH definition when validated by CALIOP data. The comparison between the derived AOCH for the three passive sensors and the CALIOP AOCH is depicted in Fig. 6, with the statistics provided in Tables 2 and 3. Both EPIC and TROPOMI show higher AOCH values compared to CALIOP, with a bias of 0.8 km for both sensors. Additionally, the RMSE values for EPIC and TROPOMI are 1.25 and 1.31 km, respectively. In contrast, GEMS shows a minimal bias accompanied by a lower RMSE of 0.75. Despite the overestimation observed in EPIC and TROPOMI, their correlations with CALIOP are notably high ($R = 0.75$ and $R = 0.71$, respectively), while GEMS exhibits a slightly lower correlation ($R = 0.64$). When valid data are available from all the retrievals, all passive sensors show a notably high correlation with the CALIOP AOCH ($R > 0.9$). Specifically, GEMS demonstrates the lowest RMSE (0.38 km), while EPIC and TROPOMI show larger RMSE values of 1.54 and 1.11 km, respectively, with a tendency to overestimate ALH (Fig. S6). The major contributions to the overestimations observed in EPIC and TROPOMI come from the smoke cases over Southeast Asia (B4, B5, and B6). This suggests a potential issue with the smoke aerosol model in the EPIC and TROPOMI AOCH algorithms when applied over Southeast Asia, which warrants further investigation. Furthermore, AOCH sensitivity to changing surface reflectance and SSA is shown in Fig. S4.

The GEMS AEH algorithm retrieves both absorbing and non-absorbing aerosols, resulting in a larger dataset that is available for comparison. In contrast, EPIC and TROPOMI

exclusively retrieve AOCH for absorbing aerosols, which are determined based on the UVAI (e.g., UVAI > 1 for TROPOMI and UVAI > 1.5 for EPIC). It is therefore desirable to assess the GEMS AEH retrieval accuracy with different aerosol characteristics. We categorize GEMS aerosol retrievals into two groups using a GEMS UVAI threshold of 3 (UVAI < 3 and UVAI ≥ 3) in the subsequent analyses (Fig. 6c–d). The overall agreement between GEMS and CALIOP is better for aerosols with UVAI ≥ 3 than those with UVAI < 3, particularly for the dust cases, as evidenced by a higher $R$ value (0.75 compared to 0.42) and a lower RMSE (0.33 compared to 0.89). The improved performance for UVAI ≥ 3 can be attributed to the stronger signals of aerosol layers detected in the $O_2$–$O_2$ absorption band. Furthermore, regardless of UVAI values, as observed in Fig. 6b and Table 3, the mean bias of the GEMS AOCH tends to be higher, with a value of 0.2 km in the smoke cases compared to 0 km in the dust cases.

Based on a 2 % measurement uncertainty for the EPIC DOAS ratios in Geogdzhayev and Marshak (2018), the theoretical AOCH retrieval error is shown to remain below 1.25 km for vegetated surfaces when AOCH exceeds 1 km (Xu et al., 2019). Our analysis shows that the RMSE from all the error sources, including the measurement and retrieval uncertainties, evaluated between EPIC and CALIOP is approximately 1.25 km, which aligns with the retrieval error. The TROPOMI AOCH algorithm builds upon the framework established by the EPIC algorithm (Xu et al., 2019), with some adjustments for TROPOMI. The measurement uncertainty for TROPOMI is estimated to be 1 %–2 % (Kleipool et al., 2018). In addition to instrument errors, the TROPOMI AOCH algorithm incorporates the convolution of TROPOMI spectral data, introducing potential additional uncertainty. Our study indicates a RMSE of 1.31 km for all error sources for the TROPOMI AOCH. Assuming a retrieval error similar to EPIC, this uncertainty appears reasonable. The GEMS AEH algorithm originates from Park et al. (2016), who performed an error analysis for OMI. The instrument error was shown to be less than 10 m, stemming from a spectral wavelength error of 0.02 nm, with the total error ranging from 739 to 1276 m depending on the aerosol types. Meanwhile, GEMS has a spectral calibration accuracy of 0.002 nm (Kang et al., 2020). Our study demonstrates a RMSE of the GEMS ALH of 0.75 km, which falls within the theoretical retrieval error.

## 3.4  Passive ALH intercomparison

Upon resampling the GEMS products to match the spatial resolution of TROPOMI and EPIC, we synchronize the observation times through linear interpolation of the hourly GEMS products to facilitate a pixel-by-pixel comparison for all the ALH products. To address the possible discrepancies stemming from the different ALH definitions mentioned in Sect. 2.2, the GEMS AEH is converted to align with the EPIC

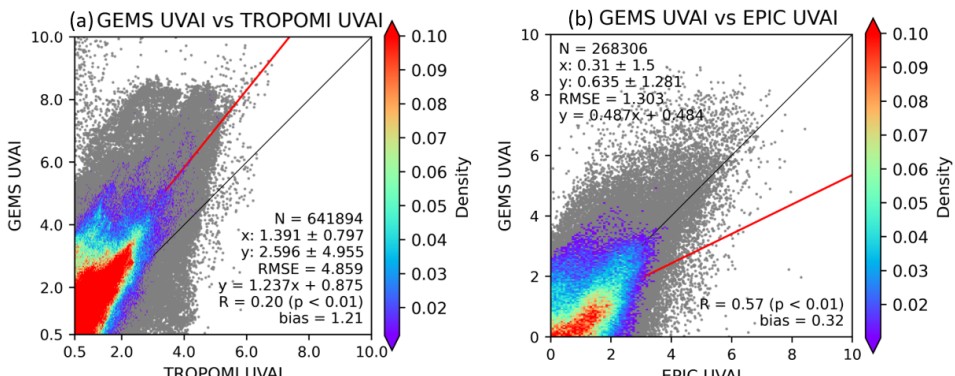

**Figure 5.** Comparison of the GEMS UVAI and the corresponding TROPOMI and EPIC products. Scatter density plots illustrate **(a)** the GEMS UVAI vs. the TROPOMI UVAI and **(b)** the GEMS UVAI vs. the EPIC UVAI. The 1 : 1 line is represented by a black solid line, while the regression line is shown in red. Grey points indicate points where the data density is less than 0.01. The dust and smoke cases are combined due to their similarity. Note that the TROPOMI UVAI does not include retrieval values below 0.5, and thus the axis begins at a minimum value of 0.5.

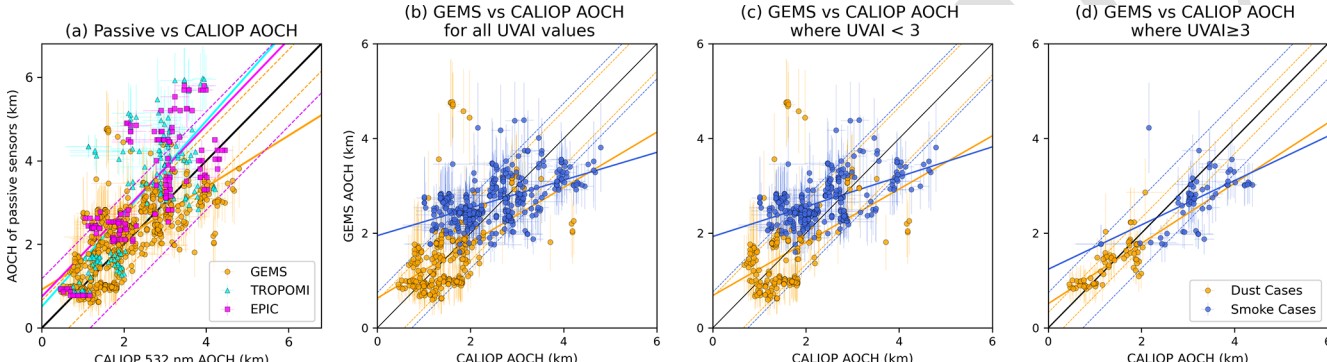

**Figure 6.** Comparison of the GEMS, TROPOMI, and EPIC AOCH with the CALIOP AOCH for all the cases. **(a)** Scatterplot of the GEMS (orange), TROPOMI (cyan), and EPIC (magenta) AOCH versus the CALIOP AOCH. The black solid line indicates the 1 : 1 line, and the dotted lines represent the error envelope within which data points for each passive satellite product fall within 1 standard deviation. Panels **(b)**–**(d)** are scatterplots of GEMS versus the CALIOP AOCH: panel **(b)** includes all the data points, panel **(c)** shows the AOCH where the GEMS UVAI < 3, and panel **(d)** shows the AOCH where the GEMS UVAI ≥ 3. For panels **(b)**–**(d)**, the orange dots represent the dust cases and the blue dots represent the smoke cases.

and TROPOMI AOCH definition. Furthermore, we categorize GEMS aerosol retrievals into two groups (UVAI < 3 and UVAI ≥ 3), similar to the analyses in Sect. 3.3.

The results of the ALH intercomparison for both the dust and smoke cases are given in Fig. 7. The GEMS AOCH exhibits a narrower range compared to TROPOMI and EPIC, which can be attributed to the different range limits used in their algorithm LUT. GEMS only allows AEH to vary within the range from 0.2 to 5 km (Park et al., 2023), while the EPIC and TROPOMI AOCH ranges from 0 to 9 km. Moreover, GEMS exhibits negative mean differences of −0.25 and −0.62 when compared to the EPIC and TROPOMI AOCH observed across all the dust and smoke cases and across both UVAI classifications. It is observed that aerosols with UVAI ≥ 3 exhibit a stronger correlation with the GEMS AOCH compared with aerosols with UVAI < 3. This can be attributed to the $O_2$ or $O_2$–$O_2$ absorption band being more

sensitive to aerosols with higher UVAI values. Setting the UVAI threshold to 4 enhances the statistical performance for UVAI ≥ 4, with increases in correlation coefficients from 0.48 to 0.61 for TROPOMI and from 0.39 to 0.46 for EPIC (Fig. S7).

Although our study cases show no CALIOP AOCH values above 5 km (Fig. 6), TROPOMI and EPIC AOCH retrievals indicate values exceeding this altitude. This is a combination of inaccurate cloud detection and the inherent sensitivity in the retrieval process of TROPOMI and EPIC. While some areas are influenced by cloud contamination, most of the high AOCH areas tend to have low AOD and are not influenced by clouds (Figs. S9 and S11). Since AOCH is more sensitive to a higher AOD (Xu et al., 2017), there is greater uncertainty in AOCH retrievals in regions with a lower AOD. Furthermore, these high AOCH areas do not show high UVAI values greater than 4; instead, they show values around 1–2, using

**Table 2.** Comparison of the GEMS, TROPOMI, and EPIC AOCH with the CALIOP AOCH in Fig. 6a.

|         | N   | EE (km) | RMSE (km) | Regression equation | x | y | R |
|---------|-----|---------|-----------|---------------------|---|---|---|
| GEMS    | 643 | ±0.7    | 0.75      | $y = 0.614x + 0.918$ | $2.1 \pm 0.9$ | $2.2 \pm 0.9$ | 0.64 |
| EPIC    | 165 | ±1.2    | 1.25      | $y = 1.025x + 0.749$ | $2.5 \pm 1.1$ | $3.3 \pm 1.4$ | 0.75 |
| TROPOMI | 144 | ±1.2    | 1.31      | $y = 1.111x + 0.510$ | $2.4 \pm 1.0$ | $3.2 \pm 1.5$ | 0.71 |

**Table 3.** Comparison of GEMS with the CALIOP AOCH for the dust and smoke cases in Fig. 6b–d.

|         | Dust cases | | | | | Smoke cases | | | | |
|---------|-----|------|-----------|-----------|---------------------|-----|------|-----------|-----------|---------------------|
|         | N   | R    | RMSE (km) | Bias (km) | Regression equation | N   | R    | RMSE (km) | Bias (km) | Regression equation |
| All       | 267 | 0.46 | 0.771 | 0.00  | $y = 0.684x + 0.63$  | 376 | 0.52 | 0.707 | 0.20  | $y = 0.293x + 1.949$ |
| UVAI < 3  | 193 | 0.42 | 0.884 | −0.01 | $y = 0.562x + 0.684$ | 308 | 0.55 | 0.656 | 0.34  | $y = 0.315x + 1.931$ |
| UVAI ≥ 3  | 74  | 0.75 | 0.328 | 0.03  | $y = 0.636x + 0.511$ | 68  | 0.57 | 0.542 | −0.46 | $y = 0.468x + 1.239$ |

the GEMS UVAI as a reference. Therefore, both cloud detection inaccuracies and the low sensitivity of AOCH retrieval to low AOD contribute to the observed high AOCH in these areas, with the latter being more dominant in our selected cases.

## 3.5 Diurnal variation of the GEMS and EPIC ALH

We present a comparison of the diurnal variations of ALH in GEMS hourly observations with near-hourly EPIC measurements, which provide between two and six daily observations within our region of interest. Our study domain encompasses a wide geographical area, with selected cases spanning from March to August, introducing seasonal changes that result in significant shifts in the Sun's position. Therefore, we define the relative local solar noon time for a given day as the moment when the solar zenith angle at a particular location reaches its minimum value. Using this relative local solar noon as a reference, we adjust the observation times of the GEMS and EPIC products using the relative local solar time (LST). Additionally, the GEMS AEH was converted according to the AOCH definition. Figure 8 illustrates the hourly diurnal variations of the GEMS and EPIC AOCH.

The diurnal pattern of EPIC AOCH values reveals a notable ascent in the morning, starting from heights below 3 km around 07:00–08:00 LST and peaking at approximately 4.5 km during 11:00–12:00 LST, followed by a marginal decline to approximately 3.5 km at 14:00–15:00 LST. Conversely, the GEMS AOCH remains relatively stable (around 4 km) until a more pronounced descent occurs after 10:00 LST, reaching less than 3 km at 14:00–15:00 LST. Note that, most of the time, GEMS AOCH values are lower compared to EPIC. The diurnal variation of AOCH reveals a slightly different pattern when the UVAI exceeds 3 (Fig. 8b). The GEMS AOCH when UVAI > 3 shows a gradual increase

until around 10:00–11:00 LST, followed by a notable decline, consistently maintaining lower AOCH values compared to the dataset, inclusive of all UVAI values. Conversely, the EPIC AOCH remains relatively steady until it experiences a rise between 10:00 and 14:00 LST and subsequently declines at 14:00–15:00 LST. There are notable differences in the AOCH diurnal variation between GEMS and EPIC, with GEMS showing a significant decrease throughout the daytime, whereas EPIC shows a gradual increase followed by a subsequent drop.

Previous studies indicate that ALH tends to be higher in the afternoon, a phenomenon that may be partially affected by the planetary boundary layer height (PBLH). Xu et al. (2017) found a higher EPIC AOCH in the afternoon, possibly indicating a relationship with the diurnal evolution of tropospheric convection. Lee et al. (2019) observed that aerosol heights tend to rise in the afternoon and early evening, likely due to the development of the boundary layer's mixed layer. Lu et al. (2023) conjectured that the diurnal cycle (ascending in the morning and descending in the afternoon) of Saharan dust plume height is a consequence of the diurnal variation of solar heating, which leads to thermal buoyancy lifting of the dust layer, combined with the diurnal evolution of the boundary layer. Unfortunately, our dataset lacks sufficient information after 15:00 LST, making it challenging to discern the relationship between AOCH and the PBLH, particularly when the PBL collapses in the late afternoon. Therefore, we examined 3 years of Modern-Era Retrospective Analysis for Research and Applications Version 2 (MERRA-2) data from 2021 to 2023, focusing on March within the East Asian domain. The MERRA-2 AOCH is defined like the CALIOP AOCH in the paper, weighted by the optical depth at each vertical layer using aerosol extinction vertical profiles. A similar diurnal variation is observed between the PBLH and the MERRA-2 AOCH calculated with

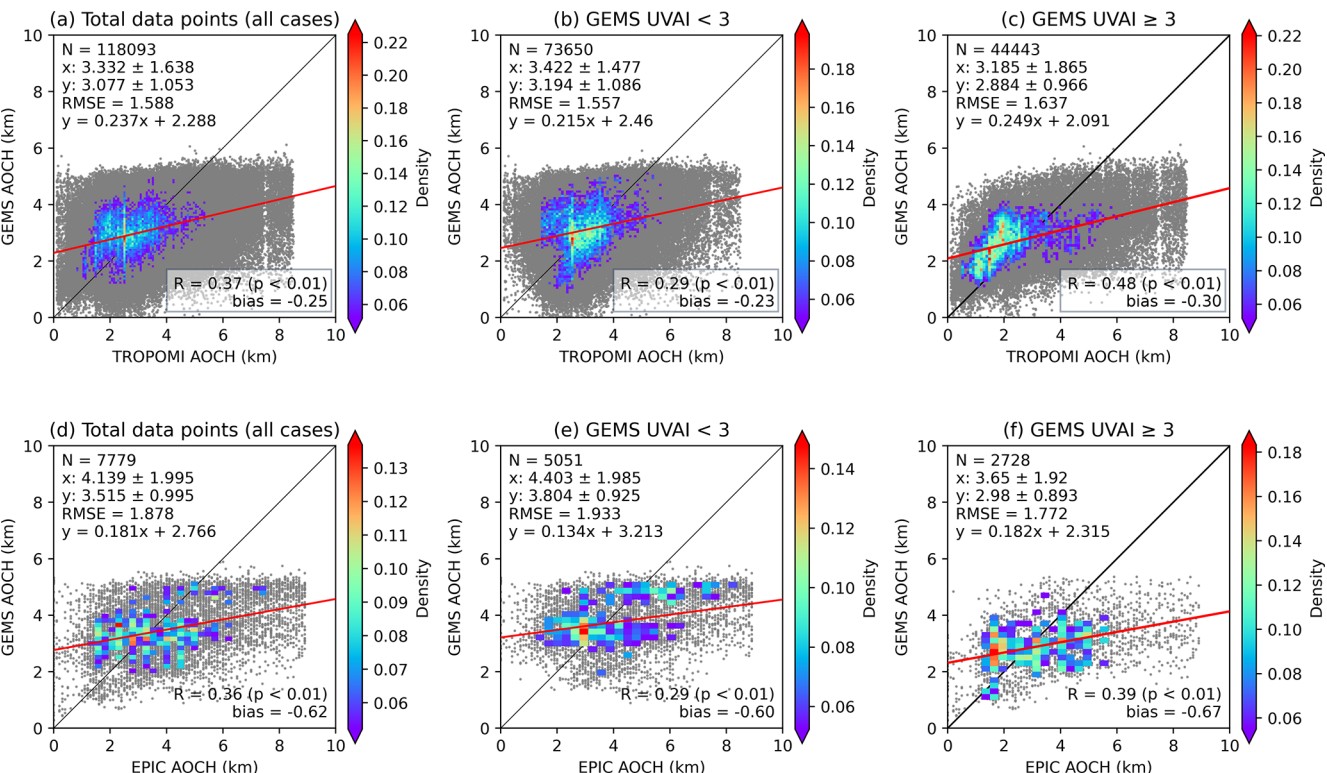

**Figure 7.** Intercomparison of AOCH values from GEMS, TROPOMI, and EPIC for all cases (dust and smoke combined) as a function of the UVAI. The density scatterplots show the AOCH comparison between GEMS and TROPOMI **(a–c)** and between GEMS and EPIC **(d–f)**. Panels **(b)** and **(e)** represent GEMS data for UVAI < 3, while panels **(e)** and **(f)** represent data for UVAI ≥ 3. GEMS AEH values have been converted to align with the AOCH definitions used by EPIC and TROPOMI.

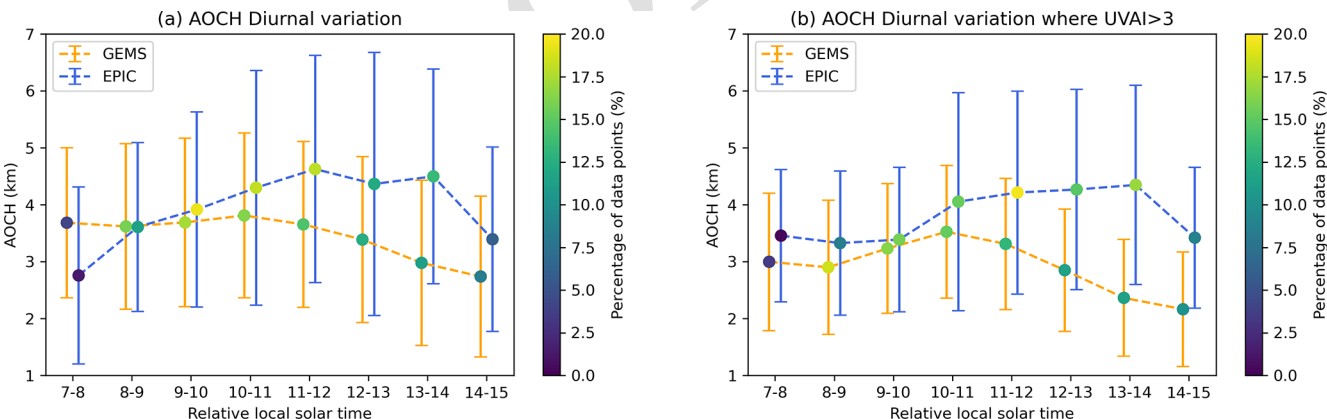

**Figure 8.** Diurnal variation of AOCH between GEMS and EPIC at relative local solar times. **(a)** AOCH diurnal variation for all pixels. **(b)** Same as **(a)** but only for UVAI > 3. Yellow lines indicate the GEMS AOCH error bar (standard deviation), blue indicates the corresponding EPIC AOCH, and dots represent the percentage of data points during each hour. The GEMS AEH has been adjusted to match the EPIC AOCH definition.

the aerosol extinction below the PBLH (Fig. S8). This indicates that, when most aerosols are located within the PBL, the diurnal variation is affected by the PBL process, which changes the PBLH during the daytime. The diurnal variation of the EPIC AOCH from Fig. 8 is consistent with the MERRA-2 PBLH and AOCH calculated by extinction below

the PBL, ascending throughout the morning and descending after 14:00 local time (LT), although the EPIC AOCH values are higher than the MERRA-2 AOCH due to its constraint by the PBLH. The diurnal variation of the GEMS and MERRA-2 AOCH shows similarities for the afternoon decrease. However, the GEMS AOCH, which shows an overall decrease

throughout the day, does not coincide with the MERRA-2 data, which show an increase until 14:00 LT. Despite the comparison of the ALH diurnal variation from satellite observations and model reanalysis, the validation of the diurnal variation of ALH still remains a significant challenge due to a lack of spatially and temporally resolved active remote sensing measurements. In addition, for passive remote sensing, potential artifacts such as scattering angle bias for geostationary satellites and contamination of cloud edges may influence the diurnal cycle of aerosol height. Additionally, the limited number of data points obtained from the selected case study dates could introduce uncertainty when attempting to generalize the ALH diurnal cycle.

## 4   Case study

We present a detailed analysis of GEMS, EPIC, and TROPOMI ALH retrievals during transport for a dust plume (D1) and a smoke plume (B6). Diurnal variations of ALH from GEMS and EPIC for the dust or smoke cases are also discussed.

### 4.1   Dust plume case

Figure 9 shows GEMS, TROPOMI, and EPIC ALH retrievals for a selected dust case on 28 March 2021 (D1). The GEMS AEH was adjusted to the EPIC/TROPOMI AOCH definition for consistent comparison. The first column presents the GEMS AOCH, with magenta lines depicting the CALIOP ground track over the GEMS map at the closest time of CALIOP measurement, and the second column shows the EPIC and TROPOMI AOCH aligned with the closest GEMS measurement time. The AOD and UVAI maps for all the satellites are shown in Fig. S9. This case is a spring dust event originating from the Gobi a few days before reaching China on 28 March 2021, specifically near Beijing, as indicated by the red star in the middle of the research domain (Fig. 9a). In the dust plume area, the GEMS AOCH peaks at high values ($\sim$ 3 km) at 01:45 and 02:45 UTC (Fig. 9a–b) before gradually decreasing to $\sim$ 1.5 km by 06:45 UTC (Fig. 9f). In contrast, the EPIC and TROPOMI AOCHs maintain relatively consistent values at 1–2 km. For this dust case, hourly GEMS observations reveal clear hourly variations, while characterizing diurnal changes from TROPOMI and EPIC is challenging due to their limited number of observations compared to GEMS.

After converting all passive ALH products according to the CALIOP AOCH definition, the comparison of the GEMS, TROPOMI, and EPIC AOCH with the CALIOP AOCH for this dust case was conducted and is shown in Fig. 10. GEMS has the greatest number of data points due to its valid retrievals for both scattering and absorbing aerosols and its high spatial resolution. For this specific case, the EPIC AOCH shows the largest correlation coefficient of all

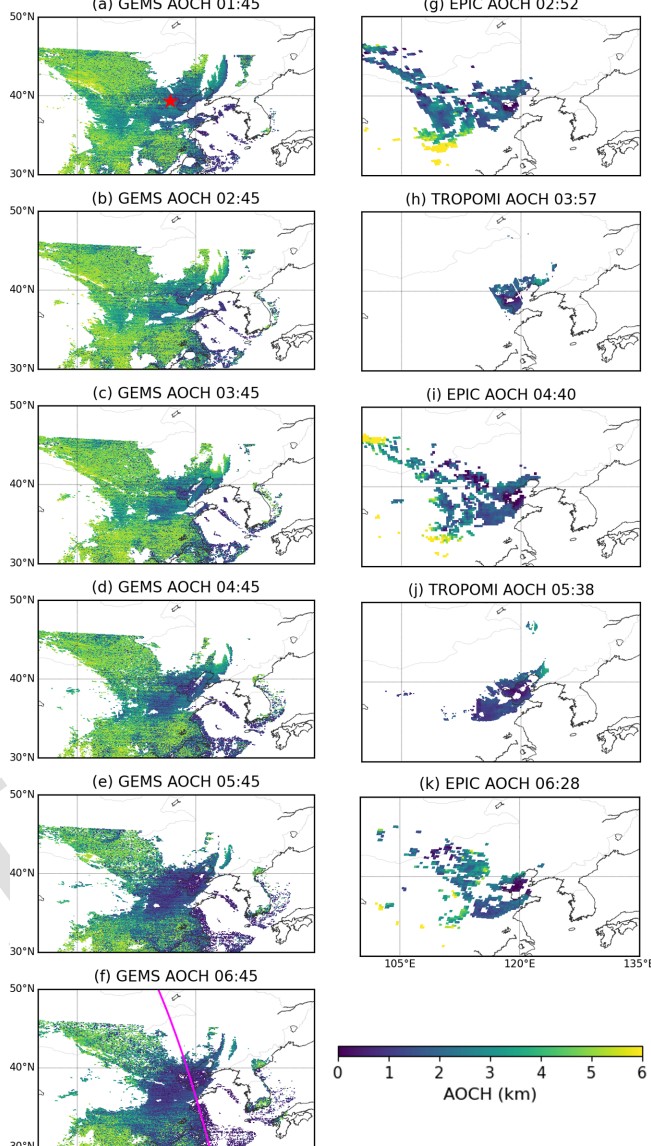

**Figure 9.** The first column **(a–f)** shows the hourly GEMS AOCH (GEMS AEH adjusted to the EPIC/TROPOMI AOCH definition) from **(a)** 01:45 UTC to **(f)** 06:45 UTC. **(g–k)** The EPIC and TROPOMI AOCH aligned with the nearest GEMS observation times for a dust plume event on 28 March 2021. The magenta line on the GEMS maps in the first column indicates the CALIOP ground tracks, which have the closest observation times with GEMS. The red star indicates the dust plume area near Beijing.

of them ($R = 0.76$), and the TROPOMI AOCH also has a high correlation coefficient ($R = 0.6$) and the lowest RMSE of 0.33 km. Although CALIOP can capture multiple layers of aerosols from extinction coefficients, the passive sensors used in this study assume a single vertical profile, thereby retrieving AOCH where a stronger signal is detected. In Fig. 10b, CALIOP identifies discontinuous high extinction coefficients at 38° N latitude and 119.5° E longitude, leading

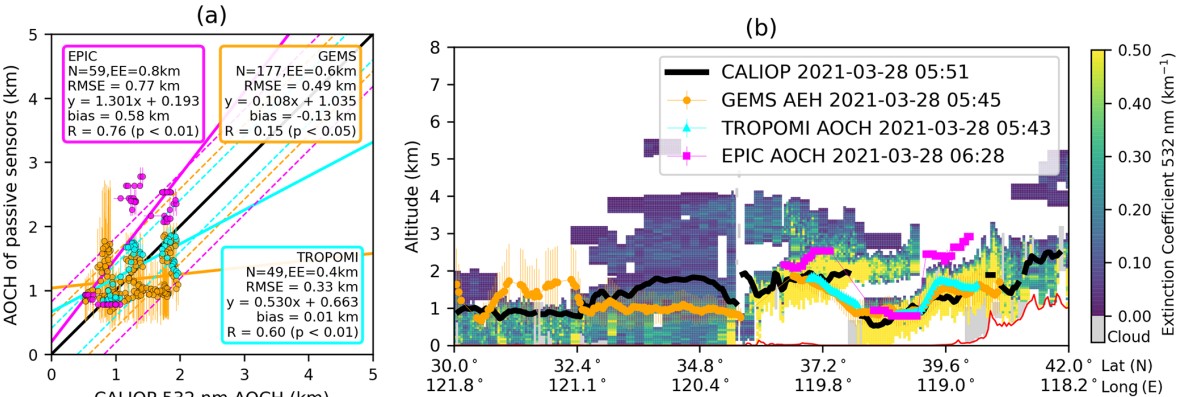

**Figure 10.** Comparison of the GEMS, TROPOMI, and EPIC AOCH with the CALIOP AOCH for a dust case over East Asia on 28 March 2021. **(a)** Scatterplot of GEMS (orange), TROPOMI (cyan), and EPIC (magenta) versus the CALIOP AOCH. **(b)** The GEMS, TROPOMI, and EPIC AOCH in the vertical profile of the CALIOP aerosol extinction curtain plot. All the ALH products are converted according to the CALIOP AOCH definition.

to a discontinuous CALIOP AOCH. While the EPIC AOCH shows a discontinuity from the absence of the retrieval in between the two layers, GEMS and TROPOMI exhibit continuous retrievals that consistently follow the stronger signal. Consequently, discrepancies between CALIOP and passive sensors may be more pronounced in the presence of multiple aerosol layers. Further investigation is needed for a comprehensive study of multilayer aerosol plumes.

Figure 11 shows the regional averaged ALH during this dust plume transport from 27 to 29 March 2021. To focus on the consistent area covered by the thickest dust plume, different UVAI thresholds were selected empirically. Pixels where UVAI values of their own products are higher than 3, 1, and 2 were considered for GEMS, TROPOMI, and EPIC, respectively. For CALIOP, collocated pixels along the track with GEMS UVAI > 3 were considered. Maps of the UVAI and the regional ALH for all the products are provided in Figs. S9 and S10. The mean AOCH values of the dust plume from all the products show good agreement, falling within a reasonable error range of < 1 km. GEMS measurements show that the dust plume is located at 4–5 km on 27 March, descends to ∼ 3 km on 28 March, and remains there until 29 March, which is consistent with the EPIC and TROPOMI measurements. These daily changes in ALH reflect the atmospheric subsidence of dust aerosols during transport. Although the daily mean of the AOCH values changes during the plume transport, GEMS shows a similar diurnal variation each day, increasing in the early morning and decreasing consistently throughout the daytime.

### 4.2 Smoke plume case

Figure 12 displays ALH retrievals from GEMS with TROPOMI and EPIC for one of the selected smoke cases on 17 April 2023 (B6). The first column displays the GEMS AOCH, which was converted from the GEMS AEH to match

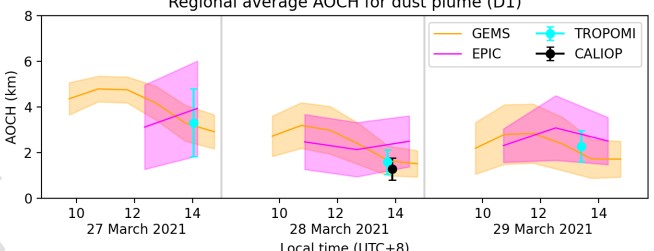

**Figure 11.** Time series plot of the regional averaged AOCH for a dust plume on 28 March 2021 (D1). The AOCH of GEMS, TROPOMI, and EPIC is represented in orange, cyan, and magenta, respectively. The lines and shadows indicate the mean and standard deviation, respectively. The CALIOP AOCH is represented by the black error bar. The GEMS AEH, EPIC AOCH, and TROPOMI AOCH are all converted according to the CALIOP AOCH definition.

the EPIC/TROPOMI AOCH definition. The second column shows the EPIC and TROPOMI AOCH aligned with the closest GEMS measurement time. In addition, AOD and UVAI maps are provided in Fig. S11. This particular case is a smoke event in Southeast Asia, with the identified smoke plume situated predominantly over the northern areas of Laos and Thailand, as shown in the central part of the domain (20° N, 100° E) and indicated by the black star in Fig. 12a. In recent decades, the air quality in Southeast Asia has been affected periodically by a transboundary smoke and haze issue primarily linked to slash-and-burn agriculture and land-clearing practices, particularly during the dry season (Shi et al., 2014; Chang and Song, 2010). Focusing on the smoke plume that can be identified from EPIC/TROPOMI AOCH retrieval for absorbing aerosols, the GEMS AOCH ranges from 3 to 5 km, while EPIC and TROPOMI consistently show values predominantly exceeding 4 km over land. The decrease in EPIC

AOCH spatial coverage throughout the day, coupled with an increase in AOCH values (Fig. 12i, j), indicates the dissipation process of the smoke plume.

Figure 13 presents a comparison with the CALIOP AOCH, specifically highlighting the northern regions of Laos and Thailand where the smoke plume is detected along the CALIOP ground track. GEMS AEH and EPIC/TROPOMI AOCH values have been converted according to the CALIOP AOCH definition. The GEMS and CALIOP AOCH values show comparability in the range 2–4 km, as evidenced by a smaller RMSE of 0.78 km. By contrast, the EPIC and TROPOMI ALH values are approximately 2 km higher than those of the CALIOP ALH, yet they display similar vertical distribution patterns of the smoke plume over the region of 19–20° N. This suggests that the EPIC and TROPOMI AOCH retrievals exhibit a systematic positive bias for aerosols over Southeast Asia, indicating the potential need for tuning in the related smoke model, including surface reflectance and aerosol properties like size distribution, refractive index, and single scattering albedo. In general, GEMS demonstrates comparability with the CALIOP AOCH, whereas both the EPIC and TROPOMI ALHs tend to overestimate.

In Fig. 14, we present the regional averaged ALH for absorbing aerosols for this smoke case. UVAI thresholds are chosen to facilitate the comparison of ALH among GEMS, EPIC, and TROPOMI, ensuring a consistent focus on regions with comparable coverage of absorbing aerosols. The UVAI thresholds for GEMS, TROPOMI, and EPIC are set to 3, 1.5, and 2, respectively. Detailed regional ALH maps for all the products are provided in Fig. S12. Notably, since the CALIOP product is included, all GEMS AEH and EPIC/TROPOMI AOCH values have been converted according to the CALIOP AOCH definition. In contrast to the dust case discussed in Sect. 4.1, this smoke plume shows little diurnal variation throughout the day according to GEMS and an overall slight increase observed by EPIC. After 12:00 LT, GEMS and EPIC exhibit similar patterns, yet EPIC consistently registers ALH values approximately 2 km higher throughout the day. While the observation times of CALIOP do not align with the consecutive data of GEMS, the regional average of the CALIOP AOCH falls within the range of the GEMS AOCH. Additionally, the regional mean of the TROPOMI AOCH is higher than that of GEMS but lower than EPIC for this smoke plume.

## 5 Conclusion and discussion

Aerosol vertical distribution is important for assessing the aerosol climate impact, surface air quality, and remote sensing of aerosols. In this study, we compared multiple ALH products from the satellite platforms of GEMS, EPIC, and TROPOMI that use oxygen (or oxygen-dimer) absorption bands, specifically the $O_2$–$O_2$ band at 477 nm for GEMS and

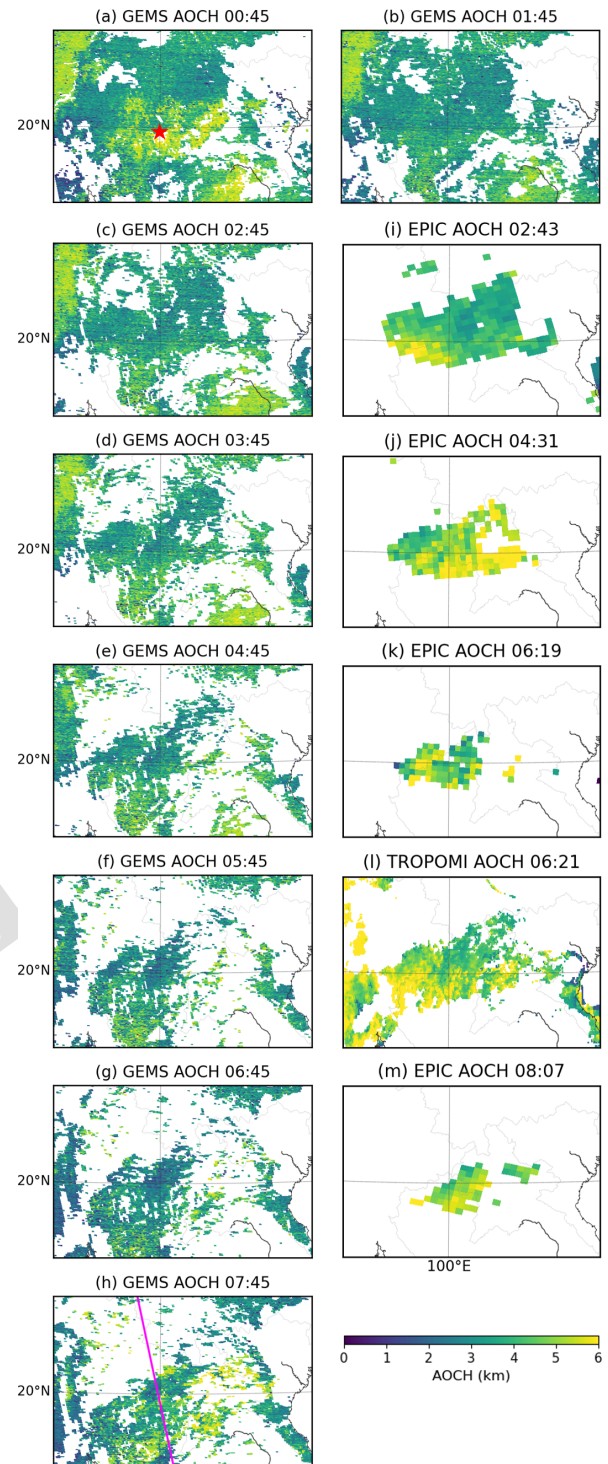

**Figure 12.** The first column, along with the first row of the second column **(a–f)**, shows the hourly GEMS AOCH (GEMS AEH adjusted to the EPIC/TROPOMI AOCH definition) from **(a)** 00:45 UTC to **(h)** 07:45 UTC. **(i–m)** EPIC and TROPOMI AOCH aligned with the nearest GEMS measurement times for a smoke plume event on 17 April 2023. The magenta line on the GEMS maps in the first column indicates the CALIOP ground tracks, which have the closest observation times with GEMS. The red star indicates the smoke plume area.

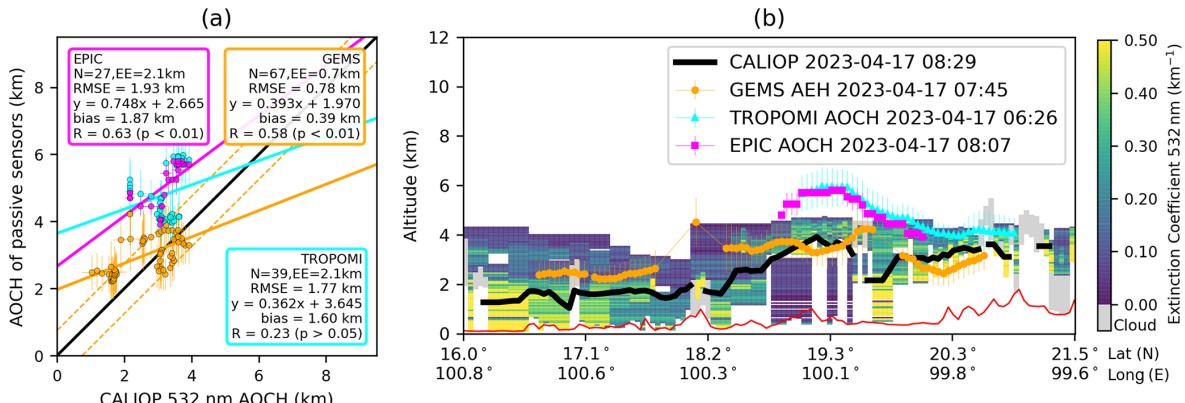

**Figure 13.** Same as Fig. 10 but for a smoke case over Southeast Asia on 17 April 2023.

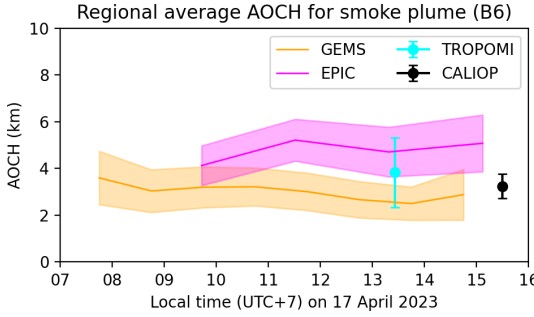

**Figure 14.** Time series plot of the regional averaged AOCH for a smoke plume on 17 April 2023. The AOCH of GEMS, TROPOMI, and EPIC is represented in orange, cyan, and magenta, respectively. The lines and shadows indicate the mean and standard deviation, respectively. The CALIOP AOCH is represented by the black error bar. The GEMS AEH, EPIC AOCH, and TROPOMI AOCH are all converted according to the definition of the CALIOP AOCH.

the $O_2$ A and B bands for TROPOMI and EPIC. Several dust and smoke plume cases over different regions in Asia covered by the GEMS field being studied were selected for comparison. Adjustments have been made to account for the inherent variations in the definitions of ALH among different products, ensuring an apple-to-apple comparison.

As part of the ALH retrieval evaluation, we also evaluated the AOD retrievals from GEMS, EPIC, and TROPOMI with the AERONET AOD and compared the UVAI of these satellite platforms. Compared with AERONET, the GEMS AOD at 443 nm demonstrates a strong positive correlation in both the dust ($R = 0.9$) and smoke ($R = 0.88$) cases. Discrepancies arise at 680 nm for the dust cases, indicating potential inaccuracies in the GEMS dust aerosol model. TROPOMI and EPIC tend to overestimate AOD in the dust cases due to underestimated surface reflectance. The inaccuracies in the GEMS dust aerosol model contribute to the significant differences in the GEMS AOD compared to TROPOMI and EPIC. Additionally, the differences are compounded by other

causes, including potential inaccuracies in the surface reflectance in TROPOMI and EPIC. In addition, the GEMS UVAI is consistently larger than the TROPOMI UVAI by 1.2, whereas it shows better agreement with the EPIC UVAI, with a smaller bias of 0.32.

The results indicate that EPIC and TROPOMI generally overestimate ALH values compared to CALIOP, with RMSE values of 1.25 and 1.31 km, respectively. In contrast, GEMS demonstrates a minimal bias and the lowest RMSE of 0.75 km, although it exhibits a slightly lower correlation with CALIOP ($R = 0.64$). When valid data from all the sensors are present, higher correlations ($R > 0.9$) with the CALIOP ALH are observed across all the passive sensors. However, TROPOMI and EPIC still tend to overestimate the ALH. Categorizing GEMS aerosol retrievals based on a UVAI threshold of 3 revealed better overall agreement with the CALIOP ALH for aerosols with UVAI $\geq 3$. While comparing GEMS with EPIC and TROPOMI, a narrower range in the GEMS AEH was seen, due in part to limitations in its algorithm LUT within the range of 0.2 to 5 km, while EPIC and TROPOMI had LUTs extending from 0 to 9 km.

The diurnal pattern of EPIC AOCH values shows a morning ascent ($\sim 4.5$ km) and a descent after noon, while GEMS remains relatively stable until a more pronounced descent in the late morning and early afternoon to below 3 km. MERRA-2 data analysis indicates a similar diurnal variation between the PBLH and MERRA-2 AOCH suggesting that, when most aerosols are within the PBL, their diurnal variation is affected by the PBL process. The validation of the diurnal variation of ALH is very challenging due to the limited spatial and temporal resolution in active remote sensing measurements and potential artifacts in passive remote sensing data.

Lastly, we presented a detailed analysis of both a dust case and a smoke case to compare differences in the spatial and temporal distribution of ALH. GEMS, with its higher temporal resolution, captured diurnal variations of the dust plume, while EPIC showed the highest correlation ($R = 0.76$) and

TROPOMI had the lowest RMSE (0.33 km) with CALIOP. In the smoke plume case, both EPIC and TROPOMI exhibited a consistent positive bias of over 1.5 km compared to CALIOP, likely due to differences in the surface reflectance and aerosol property assumptions. Overall, the passive sensors demonstrated consistent trends in ALH, but the hourly observations of GEMS provide valuable insights into the diurnal variations of ALH from individual dust and smoke plumes which the limited observations from other sensors could not fully capture.

In conclusion, our comprehensive analysis provides a thorough evaluation of the performance and comparative assessment of ALH, AOD, and UVAI retrievals from GEMS, EPIC, and TROPOMI. The comparison of the ALH definition among the different sensors highlights the need for standardization, ensuring a consistent basis for comparisons. The results from this study help enhance our understanding of aerosol plume characteristics, overcoming challenges associated with previously difficult aspects such as the comparison of ALH diurnal variations. Furthermore, we offer insights for future ALH product development by identifying and addressing the limitations in inputs from each retrieval algorithm, such as the impact of aerosol models and surface reflectance.

## Appendix A: List of abbreviations

| | |
|---|---|
| AE | Ångström exponent |
| AEH | Aerosol effective height |
| AERONET | AErosol RObotic NETwork |
| ALH | Aerosol layer height |
| AOCH | Aerosol optical central height |
| AOD | Aerosol optical depth |
| CALIOP | Cloud-Aerosol Lidar with Orthogonal Polarization |
| CALIPSO | Cloud-Aerosol Lidar and Infrared Pathfinder Satellite Observation |
| CLARS-FTS | California Laboratory for Atmospheric Remote Sensing Fourier Transform Spectrometer |
| DOAS | Differential optical absorption spectroscopy |
| DOFS | Degrees of freedom for signal |
| DSCOVR | Deep Space Climate Observatory |
| EPIC | Earth Polychromatic Imaging Camera |
| GEMS | Geostationary Environment Monitoring Spectrometer |
| HAF | Highly absorbing fine |
| IR | Infrared |
| LST | Local solar time |
| LUT | Lookup table |
| MBE | Mean bias error |
| MERRA-2 | Modern-Era Retrospective Analysis for Research and Applications Version 2 |
| MPLNET | NASA Micro-Pulse Lidar Network |
| NA | Non-absorbing |
| NIR | Near-infrared |
| OMI | Ozone Monitoring Instrument |
| PBLH | Planetary boundary layer height |
| PM | Particulate matter |
| RMSE | Root mean square error |
| SCD | Slant column density |
| SCIAMACHY | Scanning Imaging Absorption spectroMeter for Atmospheric CHartographY |
| SNR | Signal-to-noise ratio |
| SSA | Single scattering albedo |
| SWIR | Shortwave infrared |
| SZA | Solar zenith angle |
| TOA | Top of atmosphere |
| TOMS | Total Ozone Mapping Spectrometer |
| TROPOMI | TROPOspheric Monitoring Instrument |
| UV | Ultraviolet |
| UVAI | UV aerosol index |
| VisAI | Visible aerosol index |

*Code availability.* The aerosol layer height and aerosol optical depth analysis codes are available at https://doi.org/10.5281/zenodo.10408292 (Kim, 2023).

*Data availability.* The TROPOMI AOCH dataset used in this study can be found at https://doi.org/10.5281/zenodo.10407271 (Chen, 2023). The EPIC level-2 AOCH data can be found at https://doi.org/10.5067/EPIC/DSCOVR/L2_AOCH.001 (NASA/LARC/SD/ASDC, 2018a). GEMS L2 AEH V2.0 and AERAOD V2.0 can be downloaded from the National Institute of Environmental Research Environmental Satellite Center's website (https://nesc.nier.go.kr/en/html/datasvc/index.do; National Institute of Environmental Research Environmental Satellite Center, 2023). The CALIOP level-2 data were obtained from the NASA Langley Research Center Atmospheric Science Data Center and are available from https://asdc.larc.nasa.gov/data/ (last access: 15 January 2025; https://doi.org/10.5067/CALIOP/CALIPSO/CAL_

LID_L2_05kmAPro-Standard-V4-21, NASA/LARC/SD/ASDC, 2018b; https://doi.org/10.5067/CALIOP/CALIPSO/CAL_LID_ L2_05kmAPro-Standard-V4-51, NASA/LARC/SD/ASDC, 2025). Earthdata registration TS3 is required for the access to the CALIOP level-2 data.

*Supplement.* The supplement related to this article is available online at: https://doi.org/10.5194/amt-18-1-2025-supplement.

*Author contributions.* HK performed the data curation, formal analysis, and visualization and wrote most of the original draft. XC was responsible for the conceptualization, formal analysis, and methodology; for writing parts of the original draft; and for the supervision. XC also provided the TROPOMI AOCH data, and ZL provided the EPIC AOCH data. JW provided comments on the research design and supervision. MZ contributed to the formal analysis, and GRC provided comments on the writing focus and structure. SSP provided the GEMS AEH V2.0 data. All the authors, including JK, provided comments and edited the manuscript.

*Competing interests.* At least one of the (co-)authors is a member of the editorial board of *Atmospheric Measurement Techniques*. The peer-review process was guided by an independent editor, and the authors also have no other competing interests to declare.

*Special issue statement.* This article is part of the special issue "GEMS: first year in operation (AMT/ACP inter-journal SI)". It is not associated with a conference.

*Acknowledgements.* We thank the National Institute of Environmental Research of South Korea for providing the GEMS satellite data. We acknowledge the public availability of CALIOP level-2 aerosol profile data from the NASA Langley Research Center's Atmospheric Science Data Center. We thank all the principal investigators, co-principal investigators, and their staff for establishing and maintaining the AERONET sites used in this investigation.

*Financial support.* This research has been supported by the NASA EPIC/DSCOVR science team program (grant no. 80NSSC22K0503); NASA SERVIR (grant no. 80NSSC23K0244); NASA ASIA-AQ (grant no. 80NSSC23K0820); NOAA's GEO-XO-ACX concept evaluation program (grant nos. 1305M322PNRMT0542, 1305M323PNRMN0450, and NA23OAR4310303); and the Climate Program Office's Earth's Radiation Budget (ERB), Atmospheric Chemistry, Carbon Cycle, and Climate (AC4), and Climate Variability and Predictability (CVP) programs (grant nos. NA23OAR4310302, NA23OAR4310303, and NA23OAR4310304).

*Review statement.* This paper was edited by Rokjin Park and reviewed by Jeffrey Reid and three anonymous referees.

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

**Remarks from the typesetter**