# Peer review of "Aerosol layer height (ALH) retrievals from oxygen absorption bands: Intercomparison and validation among different satellite platforms, GEMS, EPIC, and TROPOMI"

_EGUsphere, 2023_

## Author Comment (AC1)

We would like to express appreciation to the reviewer for their insights and detailed review that made this paper stronger. We have taken them all into serious consideration. Our responses (in blue) for each comment (in black) and updates to the paper (in *italics*) are provided below.

Authors' response to RC1

Synopsis: This is a pretty straightforward paper comparing passive Aerosol Layer Height (ALH) retreivals generated from TROPOMI, GEMS and EPIC over two Asian domains for signciant dust and smoke events. Comparisons are made to AERONET for AOD, CALIPSO CALIOP products for the vertical centroid, and intercomparring heights between each other. Also included are comparisons of the nature of diurnal retrievals and are two example cases. As the authors note, the oxygen band retrievals are coming into their own, and while they have limited measurement degrees of freedom, they do have coverage that cannot be achieved with lidar systems. Overall, the paper is certainly appropriate to AMT and can have importance to the community. This said, I do think the paper requires major revisions. These revisions are in two prime areas.

First, they authors delicately ignore a number of sampling considerations throughout the paper. Notably, when they compare the passive products to CALIOP, implicit in that comparison is excellent viewing conditions. If there was say thin cirrus, there would not be a CALIOP retrieval to compare. For the intercomarison between retrievals without CALIOP, there clearly is a wider and higher distribution of heights. Notable are little "isolate" retrievals with very high height retrievals. The authors need to examine this data population closely. This may require a lot of hand analysis.

We aimed to capture all dust and smoke events across Asia from 2021 to 2023 during CALIOP overpasses. However, these occurrences were limited, spanning only a handful of days. We recognize the concern regarding the scarcity of data points. Throughout the paper, we have underscored our focus on "golden days" marked by ideal viewing conditions and the absence of cloud contamination.

A second concern is the reported diurnal cycle. Certainly we expect the diurnal cycle to be important to retrievals such as GEMS and EPIC that have widely varying scattering angles for retrieval physics if not reality, and thus can be important to include in a paper like this. Indeed, all of the major AOD retrievals from geostationary have significant diurnal biases. However, the authors gloss through potential artifacts to explain diurnal differences, and are highly suggestive that the diurnal cycle from EPIC (their retrieval mind you) is real. However their explanation of the nature of diurnal aerosol height is flatly wrong-they suggest that the maximum aerosol height should be around solar noon due to PBL mixing. First, mixing will sustain itself through the afternoon and thus we would not expect a solar noon peak. Second, their product uses aerosol index and they are not as sensitive to the PBL anyways. Thus physically their explanation does not make sense. As EPIC stares at the sunny hemisphere of the earth, they may have an unaccounted for diurnal scattering angle issue in their RT code that

causes symmetry along solar noon, or they may have a cosine response/resolution issue and perhaps also an associated resolution-based cloud mask bias. Now in full transparency, I am a co-author on a paper from this group that reported a strong diurnal height cycle of dust off of Africa. This said, as part of my contribution I pushed for softer language and examination of biases. As we see the exact same behavior here, I am thinking even more about diurnal bias in their EPIC retrievals. They have to come clean up front, and say they have no current way to evaluate which is correct, noting the possible artifacts, or remove the section all together. I strongly encourage the authors to look at some ground based lidar data in the region to verify their algorithm. There are plenty of Japanese and NASA lidars in this region that they can use.

Thank you for your concern. Firstly, it's important to clarify that our paper does not advocate for any specific product. Instead, we position ourselves as impartial users of several aerosol layer height products. While it's true that some of our co-authors contribute to the development of the EPIC product, we also have team members involved with GEMS and TROPOMI aerosol products. We have taken great care to ensure that our analysis remains objective and unbiased towards any particular product.

We acknowledge your valid concern regarding potential artifacts such as scattering angle issues for GEMS, however, EPIC focuses on backscattering with narrow varying scattering angles given its long distance to the Earth. Still, biases in cloud masking, particularly at lower resolutions near the edges, may influence the diurnal cycle of aerosol height. Additionally, the limited data points obtained from selected case study dates could introduce uncertainty when attempting to generalize aerosol height trends.

In this section, our intention was to illustrate the disparity in diurnal patterns between GEMS and EPIC, rather than definitively concluding that aerosol layer height peaks at local noon time. To reflect this, we have revised the language to adopt a more cautious tone, omitting the statement of a peak at local noon time and emphasizing the inherent difficulty in determining the accuracy of either dataset. Nonetheless, we believe this section remains valuable as it underscores the contrasting diurnal patterns observed between GEMS and EPIC, even if these differences may partially be attributed to measurement biases.

Additionally, we examined three years of MERRA-2 data spanning from 2021 to 2023, focusing on March within the East Asia domain. MERRA-2 AOCH is defined same as the CALIOP AOCH in the manuscript, weighted by optical depth at each vertical layer using aerosol extinction vertical profiles. Orange line indicates the average AOCH where AOCH is lower than the PBLH and the red represents the average of those PBLH values. Furthermore, AOCH calculated considering the aerosol extinction only below the PBLH (illustrated by the green line) show similar diurnal pattern with the PBLH. These similar diurnal variation between AOCH and PBLH indicates that when most aerosols locate within the PBL, diurnal variation is affected by the PBL process, which changes the PBLH during daytime.

The diurnal variation of EPIC AOCH from Figure 8 is consistent with the MERRA-2 PBLH (grey line) and AOCH calculated by extinction below the PBL (green line), as ascending throughout the morning and descending after 2 pm local time, although EPIC AOCH values are higher than MERRA-2 AOCH, due to its constrain by PBLH. The diurnal variation of GEMS and MERRA-2 AOCH show similarities for the afternoon decrease. However, GEMS showing an overall decrease throughout the day, does not coincide with the MERRA-2 data, which shows in increase until 14 local time.

[Figure]

To further support our analysis of diurnal variations, we tried exploring data from MPLNET. However, only one MPLNET site falls within our Southeast Asia domain, it does not coincide with any of the dust or smoke plumes on our selected dates. We remain open to exploring additional ground-based lidar datasets for future studies.

In addition to these two science concerns, the paper has many minor copy editing typos, language irregularities, verb tense, dropping articles etc., Just an example, starting a paragraph on line 84 "The light travels longer path when aerosols locate at lower altitude than those at higher altitude, leading to more absorption from more O2 molecules in longer path (Ding et al., 2016; Xu et al., 2019)." The paper has sentences like this in almost every paragraph, and frequently drops articles (the/a/an). I suggest the authors utilize university copy editing programs before the paper can proceed.

We have addressed grammatical errors and improved sentence structures. Your feedback is appreciated. Thank you.

Specific notes

Abstract Line 15 (And Intro lineS 72-76). I think from the beginning the authors should be a little more modest about what an aerosol layer height retrieval means. It does indeed provide information, but it is only a single degree of freedom in what can be a complex aerosol vertical structure. From the beginning, Aerosol Layer Height (ALH throughout) can be a bit misleading, and largely based on a false presupposition-that there is one layer. I am ok with language like "scale height", or "aerosol centroid", but honestly ALH has always bugged me as a labeled variable. Indeed, as noted in line 23 they have to make adjustments in definitions from the different product lines. I only ask they authors be mindful of this throughout the paper.

We appreciate this comment regarding the terminology of aerosol layer height. Given the diverse definitions and names of aerosol height products across platforms, we used the term "aerosol layer height" (ALH) to encompass all such products, including aerosol optical centroid height (AOCH), or aerosol effective height (AEH). Acknowledging the limitation of ALH in multiple aerosol layer structure, we revised some parts in the manuscript.

*(Abstract) Aerosol vertical distribution is crucial for assessing surface air quality and the impact of aerosols on climate. Although aerosol vertical structures can be complex, assuming a shape of the aerosol vertical profile enables the retrieval of a single parameter, aerosol layer height (ALH), from passive remote sensing measurements.*

*(Introduction) Hence, many algorithms have been developed to extract a single piece of information regarding aerosol vertical distribution, with a primary emphasis on aerosol layer height, which approximates the altitude of aerosols from a presumed aerosol vertical profile shape.*

Abstract line 24-25, you may want to mention how the products cross correlate.

We included the cross correlation of the products.

*When comparing GEMS ALH to TROPOMI and EPIC, GEMS shows a narrower range and correlation coefficients are both below 0.4 (R < 0.4).*

Abstract line 30 "EPIC and TROPOMI tend to overestimate AOD by 0.33 km and 0.23 km, respectively, in dust cases" I am not sure where the km fits into this, as AOD is unitless.

Thank you for pointing this out. This was a mistake. We removed 'km', since AOD is unitless.

Introduction. Line 52 "Cloud-Aerosol Lidar with Orthogonal Polarization (CALIOP) on board with the Cloud-Aerosol Lidar and Infrared Pathfinder Satellite Observation (CALIPSO) platform detects aerosol backscattering extinction profile with fine vertical resolution (Winker et al., 2013)." This is a bit of unusual language. What CALIOP measures is attenuated backscatter. It retrieves aerosol extinction profiles, but these are retrievals and can have uncertainties of their own.

We revised the sentence.

*Cloud-Aerosol Lidar with Orthogonal Polarization (CALIOP), on board with the Cloud-Aerosol Lidar and Infrared Pathfinder Satellite Observation (CALIPSO) platform, detects*

*backscatter signal from different altitudes and retrieves aerosol extinction profiles with high vertical resolution (Winker et al., 2013)*

Introduction. Line 55. "With the retirement of CALIPSO in August 2023, passive remote sensing will become the only routine technique from space for filling the data gap of measuring aerosol vertical distribution before next lidar dedicated to measure aerosols are launched into space." Technically this is not true, as the Chinese have an HSRL in space. But they don't release the data. We will see if they ever do.

Thanks for the information. We modified the sentence.

*With the retirement of CALIPSO in August 2023, passive remote sensing will become the only routine technique accessible for the public at present, from space for filling the data gap of measuring aerosol vertical distribution before next lidar dedicated to measure aerosols are launched into space.*

Methods, Section 2.3. I think throughout the paper, it needs to be emphasized that "golden days" are being used in this evaluation. I have no objection to this per say, as long as they make it clear in the abstract and introduction that these results are for ideal viewing conditions, and indeed day to day "mileage may vary considerably" Indeed, the very use of the CALIPSO as a verification dataset implies that they don't have to worry so much about things like cirrus or other cloud contamination.

We have emphasized this in the abstract and introduction.

*(Abstract) All analyses are conducted for selected "golden days", which represent ideal viewing conditions for typical dust and smoke cases.*

*(Introduction) Considering that the spatial coverage of CALIOP is limited, we carefully selected "golden" cases where dust and smoke events favor the retrievals from all three sensors. This selection can maximize the signal to noise ratio for ALH retrieval, and hence, the evaluation can shed light on the future improvement to bring the closure of various types of retrievals. Note that these conditions may differ from those observed on non-selected days.*

*(2.3 Comparison Approach) Given the availability of EPIC/TROPOMI retrievals for absorbing aerosols, we focus our comparison on a selection of "golden days" characterized by ideal viewing conditions for dust and smoke cases, excluding cloud-covered areas, as observed within GEMS field of regard from 2021 to 2023.*

Results

Line 295: I am not entirely sure how meaningful the stats are here "For dust cases, both TROPOMI and EPIC AOD exhibit a positive bias compared to AERONET AOD, with values of 0.23 and 0.33 for TROPOMI and EPIC, respectively." As these are for a distribution of AODs with different densities across AOD values. You can calculate bias as a function of AOD, or if it is linear enough a slope bias.

Thank you for your feedback. We assessed the difference in AOD between passive sensors with

AERONET AOD as a function of different AOD bins. GEMS AOD consistently underestimate AERONET AOD, with the disparity increasing as AOD levels rise. EPIC and TROPOMI show positive bias, which is consistent with Fig. 3, but no significant trend in the difference is observed across different AOD bins. We included this figure below in the Supplementary (Fig. S1).

[Figure]

Line 320-323: The authors should probably use more direct language, that based on Figure 3, the TROPOMI and EPIC dust AOD products were quite high biased (presumably as stated line 323 due to surface reflectance), and GEMS was low perhaps due to the optical model. Thus, when you cross compare, of course there is a massive bias between them. But can you also check this by looking only at data over water?

Thank you for the feedback. The figure below illustrates AOD comparisons distinguished over land and water. Comparison with AERONET data reveals that TROPOMI and EPIC AOD estimations for dust cases tend to overestimate. Specifically, in Figures (b) and (d), which compare GEMS to TROPOMI and EPIC AOD over land, we observe a similarity with AERONET comparisons: for GEMS AOD values close to zero, TROPOMI and EPIC AOD values are higher compared to GEMS. However, this trend is not seen over water (Figures (a) and (c)). The discrepancy stem from differences in surface reflectance estimation in retrieval algorithms; GEMS employs the OMI surface reflectance climatology data product OMLER v003, whereas TROPOMI and EPIC utilize climatology MODIS surface products. This suggests that surface reflectance estimates from TROPOMI and EPIC over land may require refinement for more accurate retrievals. We have included this figure in the supplementary document.

*Specifically, surface reflectance estimates for land surfaces from TROPOMI and EPIC may need refinement, as GEMS AOD values close to zero tend to be higher in TROPOMI and EPIC over land. However, this trend is not observed over water.*

[Figure]

Line 345-352: AI can be a can of worms. There are a host of other issues than those noted, including pressure assumption differences, not just land reflectance but altitude models, resolution differences resulting in different cloud effects, etc. probably you need to calculate it yourself consistently.

We acknowledge the multitude of parameters that can impact UVAI, including those you mentioned. Given the variability in UVAI across different products, we adopted a criterion for classification in our paper based on setting UVAI thresholds to achieve similar spatial coverage. To achieve this, we used TROPOMI UVAI as a reference and set the UVAI threshold accordingly, rather than directly calculating UVAI ourselves. This approach ensures consistency in spatial coverage across the datasets under comparison. We have stated this in the manuscript.

Line 354-Figure 5. Why is there no TROPOMI vs EPIC plot? I think it would be good to cross characterize everything.

A figure comparing TROPOMI versus EPIC UVAI has been included in the Supplementary (Fig. S4).

Line 383-Figure 6. Why do the axis go much further than where there is data?   Maybe set to lesser spread and put the results in a table? Also, why not do all of the passive sensors for the different AI ranges?

We reduced the axis size and put the results in two tables. We analysis AOCH for different UVAI ranges only for GEMS since TROPOMI and EPIC AOCH products are already filtered with UVAI values larger than 1 and 3, respectively, to focus on the retrieval of absorbing aerosols.

Line 408-Figure 7. It would be good to verify what is going on with AI cases bigger than 4, and what role clouds may have in increasing these values.   There are no CALIPSO verification cases with AOCH values over 4-5 km. I think it will be necessary to show that these cases where the retrieved AOCH are above such values are real, and not due to some artifact, such as thin cirrus above or unmasked low clouds below. In their case study (e.g, Figure 9) spot retrievals of very high AOCH are visible.   The authors should look in detail as to what is going on there.

We have examined the areas with high AOCH and found that the issue is a combination of inaccurate cloud detection and inherent sensitivity in the retrieval process. While some areas are influenced by cloud contamination, most of the high AOCH areas tend to have low AOD and not being influenced by clouds. Since AOCH is more sensitive to higher AOD (Xu et al., 2017), there is greater uncertainty in AOCH retrievals in regions with lower AOD. Furthermore, these high AOCH areas do not show high UVAI values greater than 4; instead, they exhibit values around 1-2, using GEMS UVAI as a reference. Therefore, both cloud detection inaccuracies and the low sensitivity of AOCH retrieval to low AOD contribute to the observed high AOCH in these areas, with the latter being more dominant in our selected cases.

Line 430-Figure 8 discussion.   The diurnal version of EPIC  looks pretty symmetrical around noon.  We have talked about this before, that this may be indicative of a retrieval bias. I don't think you can quickly dismiss this as being part of the PBL cycle, especially since the focus is on absorbing aerosol layers above the PBL.  Regardless, the effect of the PBL should maintain its height through the afternoon. The authors really need to incorporate a ground based lidar into these diurnal analyses.  In fact, they might consider dropping this section until that can be done.

We answered in one of the general comments above.

Line 485-Figure 11.  Please add a local solar time to the x-axis.

We added local time for Beijing (UTC + 8) to the x-axis.

Line 518-Figure 13(b).  The crosses for GEMS, TROPOMI and EPIC are very hard to read. The authors may want to change the color scheme and line thickness here and back on Figure 10 to make it consistent.

We changed the color scheme and line thickness. Thank you.

---

## Author Comment (AC2)

We would like to express appreciation to the reviewers for their insights and detailed review. We have taken them all into serious consideration. Our responses (in blue) for each comment (in black) and updates to the paper (in *italics*) are provided below.

Authors' response to RC2

In this study, Kim et al compared aerosol layer heights products from several satellite instruments such as GEMS, EPIC and TROPOMI using retrieval algorithms all based on oxygen absorption bands. O2-A and B bands are used for TROPOMI and EPIC, O2-O2 band is used for GEMS. To have consistent comparisons, the aerosol layer heights are converted with a similar definition. Cases studies including dust and smoke over several regions in Asia are also discussed. Discrepancies between the products of the three instruments may reveal limitations in assumed aerosol and surface models and shed new lights to improve ALH retrievals. In general, this work fits the scope of AMT, and provide detailed and thorough analysis with useful results. I have several suggestive comments which may help improve the clarity of this work.

General comments:

1. There are many acronyms, and many of them are not defined when mentioned the first time. For example: UVAI (the ultraviolet aerosol index), AEH (aerosol effective height) have been used several times, but only defined in Page 5, Line 146 and 156. I would recommend having a table defining those acronyms.

    Thank you for pointing this out. We included a list of acronyms at the end of the paper (included as appendix) and verified all acronyms when introduced for the first time in the paper.

2. Most conclusions are made by comparing with AERONET and inter-comparisons. However, each product also has uncertainties which relate to the measurement uncertainties and retrieval algorithms. I didn't find discussions on the accuracy of the measurements from these instruments, and corresponding ALH uncertainties. In principle, the ALH uncertainty from each instrument can be validated using the AERONET data too.

    We included measurement and retrieval uncertainties investigated in the level 1 calibration and the algorithm papers of each sensor and product. Then, we compared these theoretical findings with our actual validation results with CALIOP.

    *Based on a 2% measurement uncertainty for the EPIC DOAS ratios (Geogdzhayev and Marshak, 2018), the theoretical AOCH retrieval error is shown to remain below 1.25 km for vegetated surface when AOCH exceeds 1 km (Xu et al., 2019). Our analysis shows that the RMSE from all error sources, including measurement and retrieval uncertainties, evaluated between EPIC and CALIOP is approximately 1.25 km, which aligns with the retrieval error.*

    *The TROPOMI AOCH algorithm builds upon the framework established by the EPIC algorithm (Xu et al., 2019), with some adjustments for TROPOMI. Measurement uncertainty for TROPOMI is estimated to be 1 – 2 % (Kleipool et al., 2018). In addition to instrument errors, TROPOMI AOCH algorithm incorporates the convolution of TROPOMI spectral data, introducing potential additional uncertainty. Our study indicates an RMSE of TROPOMI ALH as 1.31 km. Assuming retrieval error similar to EPIC, this uncertainty appears reasonable.*

    *The GEMS AEH algorithm originates from Park et al. (2016), who performed an error analysis for OMI. The instrument error was indicated to be less than 10 m, stemming from a spectral wavelength error of 0.02 nm, with the total error ranging from 739 to 1276 m depending on aerosol types. Meanwhile GEMS has a spectral calibration accuracy of 0.002 nm (Kang et al.,*

*2020). Our study demonstrates an RMSE of GEMS ALH at 0.75 km, falling within the theoretical retrieval error.*

3.      Due to the information content, the ALH uncertainty should have strong dependency with the aerosol loading (or AOD). I don't have a clear understanding on what the AOD range is used in the discussion of ALH from this study, and how that impact the conclusion.

Thank you for your insightful comment. As AOD increases, the sensitivity of the reflectance ratio in the O2 A and B bands to AOCH also increases, as illustrated in Figure 1c-d from Xu et al. (2017). To further understand the AOD dependency of ALH uncertainty in the observational perspective, we analyzed the ALH difference between CALIOP and passive products, according to AOD bins of 0.2. The average difference between CALIOP AOCH and GEMS AOCH within each AOD bin only reaches 0.4 km. For TROPOMI, AOCH difference increase as increasing AOD, at AOD less than 1.2, with TROPOMI consistently overestimating, which resonates in Fig. 6. EPIC AOCH exhibits a linear increase in bias relative to AOD until AOD surpasses 1.2. This trend may stem from the AOD dependency within the aerosol model. We included this figure in the supplementary.

[Figure]

4.      The authors suggested that the aerosol and surface model used in the ALH retrievals may cause the discrepancy between different products. It would be useful to add more discussion on how such models impact the ALH retrievals, and how the authors would recommend to improve based on the results from this study.

We conducted a sensitivity test by modifying two key parameters: single scattering albedo (SSA) as a representative of the aerosol model, and surface reflectance as a representative of the surface model. Specifically, we increased and decreased both SSA and surface reflectance by 5% to evaluate their impact on ALH and AOD for one day of EPIC retrieval. This result in Figure below where increasing SSA and surface reflectance leads to lower AOD, but higher ALH, which is consistent with theoretical expectations. However, the complexity of the relationship between these parameters is evident, as not all data points show trends in the same direction. ALH retrieval becomes more complex due to the direct influence of SSA on ALH

estimation, compounded by the indirect impact of SSA on AOD, which also affects ALH retrieval. This underscores the complexity inherent in the relationship within each parameter, making it challenging to quantify their individual contributions.

[Figure]

Page 1, line 24 "In comparison with CALIOP ALH, both EPIC and TROPOMI ALH display a high correlation coefficient (R) higher than 0.7 and an overestimation by ~ 0.8 km, whereas GEMS ALH exhibits minimal bias (0.1 km) but a slightly lower correlation with R of 0.64."

Why there is larger bias with higher correlation? Does this indicate limitation to use correlation as a metric?

Using both correlation and bias as metric provides a more comprehensive evaluation of agreement between products, capturing both the strength of the relationship and the magnitude of differences. While GEMS has lower bias, EPIC and TROPOMI show higher correlation, indicating there may be a systematic bias that could be easily adjusted from their retrieval processes.

Page 1, line 25: UVAI not defined, what is its meaning?

Thank you for pointing this out. UVAI means ultraviolet aerosol index. We included a list of acronyms at the end of the paper (included as appendix) and verified all acronyms when introduced for the first time in the paper.

Page 2, Line 63: "…the degrees of freedom for signal (DOFS) increase from 2.1 to 2.8, which becomes sufficient for three parameter retrievals (AOD, aerosol peak height, and aerosol layer thickness)…"

How do you know that the three parameters are the right set of parameters, not other ones, such as SSA, aerosol size, etc?

This is a study from Choi et al. (2021) that focused on aerosol profiling capability. This information shows the impact of measurement factors, such as spectral resolution and coverage, SNR, radiance, and polarization on aerosol profile retrievals. The three key aerosol parameters needed to retrieve aerosol profiles are optical depth, peak height, layer thickness. While information content of other retrieval parameters including surface BRDF, aerosol microphysical properties (i.e., particle size distribution and refractive indices parameters) were also considered in their study, we focused on these three parameters that directly relate to aerosol distribution, which is the main focus of our study and Choi et al.'s work.

Choi et al. (2021) found that the degrees of freedom for signal (DOFS) for a single California Laboratory for Atmospheric Remote Sensing Fourier Transform Spectrometer (CLARS-FTS) measurement becomes sufficient to retrieve three key aerosol parameters—AOD, aerosol peak height, and aerosol layer thickness in the planetary boundary layer (PBL)—when adding a high spectral resolution (with a full-width half-maximum of 3 $cm^{-1}$ or better), polarimetric measurements with SNR of at least 212, and radiance measurements with SNR of 300 for both oxygen ($O_2$) A and $^1\Delta$ bands.

Page 4. Line 115, although O2-A, O2-B and O2-O2 bands all have sensitivities, I didn't find any discussion on the measurement uncertainties from the three sensors using those bands?

As mentioned above, we included measurement and retrieval uncertainties and compared these theoretical findings with our validation results using CALIOP.

Page 4, line 125, "Accurate retrieval of ALH requires reliable retrieval of AOD, and past studies have shown that ALH and UVAI relationship can change with AOD (Xu et al., 2017)."

Can you elaborate how the relationship will change? And how did that apply to this study?

Thank you for your insightful questions. Since UVAI depends on ALH, AOD, and SSA, its correlation with ALH changes with AOD. From Figure 12 from Xu et al. (2019), the relationship between UVAI and ALH, as well as their correlation, strengthens with increasing AOD. However, our study focuses on comparing UVAI products to select thresholds for each AOCH product that has similar spatial coverage of absorbing aerosols, as well as establishing criteria (e.g., UVAI > 3 in Figure 6,7, and 8) for comparing different AOCH products.*Accurate retrieval of ALH requires reliable retrieval of AOD since the retrieval sensitivity is strongly dependent. Furthermore, ALH algorithms use UVAI to focus on absorbing aerosols or classify aerosol types. Since different sensors have their own UVAI products, we compare these products to select appropriate thresholds for comparing AOCH.*

Page 5, Line 135, "all algorithms assume quasi-Gaussian distribution described by two parameters including centroid height and half width (fixed at 1 km) at half maxima"

It would be useful to show the formula, which can help explain what is a quasi-Gaussian distribution, and half width at half maxima. I feel FWHM (full width at half maximum) is more commonly used. (I saw the formula in later section, you may need to add a reference).

We removed terms "quasi-Gaussian distribution" and "half width" since this part is just an introduction for upcoming sections. Instead, a more detailed explanation has been included in section 2.2.

Page 5, Line 156, "aerosol types are classified by the ultraviolet aerosol index (UVAI) and visible aerosol index derived from GEMS observations"

How UVAI is used to classify aerosol types?

UVAI and VisAI are calculated using the following equation:

$$AI = -100 \left[ log \left( \frac{N_{\lambda_1}}{N_{\lambda_2}} \right)_{meas} - log \left( \frac{N_{\lambda_1}(LER_{\lambda_1})}{N_{\lambda_2}(LER_{\lambda_2})} \right)_{calc} \right]$$

In this equation, $N_{\lambda_1}$ and $N_{\lambda_2}$ represent the normalized radiances at the wavelength pairs 354/388 (477/490) nm for UVAI and VisAI, respectively. The subscripts "*meas*" and "*calc*" indicate the measured and calculated normalized radiances, respectively (Cho et al., 2024). We added the definition and meaning of UVAI in the introduction when it was first mentioned.

*Three aerosol types were classified using UVAI and the Visible Aerosol Index (VisAI), which, similar to UVAI but with visible channels, categorizes aerosols into highly absorbing fine (HAF), dust, and non-absorbing (NA) aerosols. NA aerosols are selected when UVAI yields a negative value, the dust type is determined when both UVAI and VisAI are positive, and HAF is selected when UVAI is positive but VisAI is negative (Cho et al., 2023).*

Page 5, Line 159. "For LUT generation, aerosols are assumed to be spherical and their particle size distribution, refractive index and fine mode fraction for each aerosol types are derived from global AERONET inversion climatology."

Can you confirm that whether AERONET aerosol inversion already considered non-spherical aerosols? I believe there are products used non-spherical aerosol model.

AERONET's aerosol inversion provides sphericity factor, but GEMS uses Mie theory for its LUT calculation. This limitation is because GEMS uses a computationally intensive spectral binning method, as noted by Cho et al. (2023). While spectral binning enhances stability by averaging across wavelengths and improves observation reliability by reducing random errors, it requires significant computational resources. To manage this, GEMS calculations use the Mie theory, which simplifies by ignoring the non-sphericity of dust.

*For LUT generation, aerosols are assumed to be spherical due the computationally intensive spectral binning method, as noted by Cho et al. (2023). In addition, the particle size distribution, refractive index and fine mode fraction for each aerosol type are derived from global AERONET inversion climatology.*

Page 6, Line 172, what O2AB-UI algorithm stands for?

The O2AB-UI algorithm was initially used to indicate EPIC and TROPOMI's AOCH algorithm. However, we realized use throughout the paper might lead to confusion, prompting us to remove this term entirely.

Page 6, Line 180, UVAI is defined in previous page.

Thank you for pointing this out. We defined all acronyms when introduced for the first time in the paper.

Page 6, Line 183, "only those pixels covered by lofted layer of absorbing aerosols with UVAI larger than 1.5 and AOD larger than 0.2 (at 680 nm) are analysed."

Is this the case for all following analysis? Fig 3, seems include AOD as small as 0.1 for all sensors.

This sentence applies for EPIC retrievals, and in Fig 3 (d) EPIC AOD does not have data below 0.2.

Page 6, Line 191, "However, the hyperspectral measurements from TROPOMI, unlike the EPIC measurement in narrow channels, prevent us to applying the EPIC AOCH algorithm in TROPOMI L1B data directly"

So what is the band width for TROPOMI?

TROPOMI consists of a high-resolution spectrometer system functioning across a range from ultraviolet to shortwave infrared. It incorporates seven distinct spectral bands: UV-1 (270-300nm), UV-2 (300-370nm), VIS (370-500nm), NIR-1 (685-710nm), NIR-2 (745-773nm), SWIR-1 (1590-1675nm), and SWIR-3 (2305-2385nm). However, we removed this sentence as it was unnecessary.

Page 7, Line 210, Eq (1), what is the reason not choosing a Gaussian distribution but choose the current form? If a Gaussian distribution is used, FWHM or half width at half maximum can be easily expressed by the standard deviation.

Does the choice of 1km as half width at half maximum impact the ALH results?

Thank you for a valuable comment on the quasi-Gaussian distribution. As mentioned in a previous comment, while the full width at half maximum (FWHM) is more commonly used for Gaussian distribution, in the context of our study, half width is more often used as we are following the quasi-Gaussian distribution defined by Spurr and Christi (2014). Quasi-Gaussian distribution is a generalized distribution function (GDF) profile that offers easier integration in a simple closed form without requiring error functions.

*The assumption of a 1 km half width is based on typical Lidar observations for dust and smoke aerosols, as noted by Reid et al. (2003). This same value has also been employed in the derivation of Aerosol Optical Depth (AOD) from ultraviolet (UV) observations by both TOMS (Total Ozone Mapping Spectrometer) and OMI (Ozone Monitoring Instrument), as highlighted in the work by Torres et al. (1998). Now, it is a commonly used parameter value as seen in products from EPIC, TROPOMI, and GEMS aerosol layer height retrievals.*

Xu et al. (2019) illustrated the sensitivity of the DOAS ratio ($\rho$) to the half-width parameter ($\sigma H$) in the Figure A1, showing that while the DOAS ratios exhibit a negative sensitivity to $\sigma H$ for aerosols at higher altitudes, the sensitivity becomes positive for ALH values below 1.5 km, and an error of 0.5 km in $\sigma H$ could result in a retrieval error of up to 0.3 km for ALH.

Page 8, Line 265 "Hence, the accuracy of each AOD product also influences corresponding ALH retrieval, which will be validated here by the ground-based Aerosol Robotic Network (AERONET) inversions as well."

Similar to a few previous comments, the aerosol loading itself also impacts ALH retrieval. One example can be found from polarimetric retrievals, such as Gao et al 2023, (https://doi.org/10.5194/amt-16-5863-2023). It would be useful to make it clear how AOD impacts the conclusion in this study.

We acknowledge that we should give more information on how AOD influences ALH retrieval. We included information about retrieval sensitivity of ALH to AOD as mentioned in a previous answer.

Page 8, Line 274 "Since TROPOMI and EPIC AOD products are retrieved at the wavelength of 680 nm whereas GEMS AOD products are retrieved at 354, 443, and, 550 nm,…"

Is there any estimation of the AOD accuracy derived from these bands? Which one is more accurate?

GEMS AOD is retrieved at 443 nm, with AOD at 380 nm and 550 nm subsequently calculated as part of their products. Consequently, there may be error propagation for GEMS AOD at 380 nm and 550 nm if the Angstrom exponent of aerosol types used to calculate AOD at different wavelengths is inaccurate. In contrast, TROPOMI and EPIC AOD products are retrieved at 680 nm. We clarified that the GEMS AOD is retrieved at 443 nm.

*Since TROPOMI and EPIC AOD products are retrieved at the wavelength of 680 nm whereas GEMS AOD products are retrieved at 443 nm, we estimated GEMS AOD at 680 nm from its AOD at 443nm ...*

Page 8, Line 290 "The observed underestimation of GEMS AOD at 680 nm can be in part due to an overestimation of the Angstrom Exponent (AE), which can be affected from inaccurate particle size or refractive index in the wavelength-dependent aerosol model."

What aerosol model is used?

Details regarding the aerosol models employed in each algorithm are provided in Table S1 of the supplementary document

Table S1. Aerosol model comparison used in AOD/ALH retrievals for GEMS and TROPOMI/EPIC

|  |  | HAF | Dust |
|---|---|---|---|
| mr |  | 1.46 | $0.00428 \ln \tau + 1.55$ (675 nm) |
| mi |  | 0.02044 | $0.00197 \ln \tau + 0.00268$ (675 nm) |
| Reff | Fine mode | 0.0854 | $0.0152\tau + 0.122$ |
|  | Coarse mode | 1.4115 | - |
| Veff | Fine mode | 1.5421 | $0.156\tau + 0.227$ |
|  | Coarse mode | 1.7630 | - |
| fmf |  | 0.99994 | $-0.0696 \ln \tau + 0.37$ |
| SSA |  |  | Coarse mode: $0.0214 \ln \tau + 0.949$ (675 nm) |
| Phase function |  | Mie | Fine: Mie
Coarse: Dynamic AERONET climatology |

$\tau$ is the AOD at 680 nm.

Page 11, Line 302, what is the surface model used here?

The surface estimations are included in the data section (2.1) as well as here.

*(2.1 Satellite data) The surface reflectance data used for TROPOMI/EPIC ALH retrievals involves two sources: Land surface reflectance is obtained from the MODIS surface bi-directional reflectance climatology, while water surface reflectance is derived from the GOME-2 surface Lambert-equivalent reflectivity (LER) database.*

*For TROPOMI and EPIC retrievals over land, climatological surface reflectance data from MODIS is employed. Additionally, unlike GEMS AEH retrieval algorithm that uses GEMS level 2 surface reflectance data, Cho et al. (2023) developed a new method for GEMS AOD retrieval, employing a novel hourly surface reflectance database generated through the minimum reflectance method, which integrates climatological minimum reflectance values for each pixel within a ±15-day window over a two-year period, along with monthly background AOD data. This novel surface reflectance estimation from GEMS AOD retrieval is shown effective.*

Page 12, Fig 3. "Satellite data points only with a standard deviation less than 0.3 are shown for spatial consistency."

How is the standard deviation derived?

This is because we average satellite retrievals to collocate with AERONET data. In this process of averaging for a fixed radius around AERONET sites (here we use 0.2º) we might incorrectly average spatially inconsistent satellite retrievals. Also, data points with a standard deviation exceeding 0.3 constitute approximately 3% of the total data points; consequently, we regarded them as anomalies and removed them.

Page 13 Line 318, how the dust and smoke cases are separated?

*Distinguishing between dust and smoke cases is based on the predominant aerosol type observed in the CALIOP lidar data.*

(https://www-calipso.larc.nasa.gov/products/lidar/browse_images/std_v451_index.php)

Page 13, Line 321, what dust aerosol model is used, in terms of size, refractive index etc?

Aerosol models are presented in Table S1 of the supplementary document.

Page 14, Fig 4, it seems GEMS AOD has a boundary constraint which make its less than 0.5 most of the time, at least for dust? But it seems smoke case don't have such constraint.

The AERONET validation in Fig.3 provides a more comprehensive understanding of the apparent disparity for dust cases between GEMS and EPIC/TROPOMI. Specifically, when AERONET AOD is above one, GEMS AOD tends to be lower, whereas for AERONET AOD below 1, there are many instances where GEMS AOD exceeds the one-to-one line. This indicates that a higher disparity with EPIC/TROPOMI can be expected for higher GEMS AOD values, which may appear like a boundary constraint.

Page 16, 374-375, UVAI are used to categorize GEMS aerosol retrievals. What is the meaning for UVAI? Does different category relate to different aerosol types?

UVAI computes the difference between measured and calculated (as in purely molecular atmosphere) near-UV spectral dependence (Torres et al., 1998). UVAI values approaching zero occur when the atmosphere is devoid of aerosols, or when significant non-absorbing aerosol particles and clouds are present. Aerosols capable of absorbing UV radiation, such as carbonaceous aerosols, volcanic ash, and desert dust, are the main factors leading to positive UVAI values (Torres et al., 2007). As mentioned previously, GEMS use UVAI and VisAI to classify aerosol types. TROPOMI and EPIC ALH focus solely on retrieving UV absorbing aerosols by setting a UVAI threshold, ensuring retrieval only occurs for pixels surpassing this threshold.

Page 19, Line 425, how the boundary layer height relates to the ALH? Is there any quantitative relationship?

Although we don't have results to show quantitative relationship with the boundary layer and ALH, here are some previous studies that showed the diurnal variation of aerosol layer height.

*Xu et al. (2017) found that higher aerosol layer heights (ALH) were observed in the afternoon, possibly indicating a relationship with the diurnal evolution of tropospheric convection. Lee et al. (2019) observed that aerosol heights tend to rise in the afternoon and early evening, likely due to the development of the boundary layer's mixed layer. Lu et al. (2023) conjecture that the diurnal cycle of Saharan dust plume height is a consequence of the diurnal variation in solar heating, which leads to thermal buoyancy lifting the dust layer, combined with the diurnal evolution of the boundary layer.*

Page 19, Line 443, are these UTC time? Can you also provide local time?

The time shown here is similar to local time, but constructed with the relative solar local noon times, which marks the moment when the solar zenith angle reaches its minimum value at a specific location. Using these times as the central reference point, GEMS and EPIC products for the day are adjusted to relative solar local time accordingly. This adjustment was considered necessary due to the wide geographical area covered by our study and the selected cases spanning from March to August. These months exhibit seasonal variations, resulting in notable changes in the sun's position.

Page 20, Fig 9, can you provide local time too?

Yes, we added local time for Beijing (UTC + 8) to the x-axis.

[revised manuscript text omitted]

---

## Author Response (AR2)

We would like to express great appreciation to the reviewer for their insights and detailed review that made this paper stronger and clearer. We have taken them all into serious consideration. Our responses (in blue) for each comment (in black) and updates to the paper (in *italics*) are provided below.

**Report #1 Anonymous referee #3**

The manuscript compares aerosol layer height retrievals from three different passive sensors (GEMS, TROPOMI, EPIC), using the active sensor CALIOP as a reference. In addition, the aerosol optical depth and UV aerosol index are compared between the passive sensors and AERONET observations over southeast Asia. Altogether, the text is clearly written, and the comparison methodology is scientifically sound and mostly well documented. There are few details that should be elaborated upon. For example, the text does an excellent job summarizing the different definitions for aerosol layer height used by the instruments. To ensure an apples-to-apples comparison, the text says that some of the aerosol layer heights are converted from one definition to another so that they all use the same definition (lines 258-259). More details should be added to explain how the aerosol layer height values are converted. Also, CALIOP data are used as a reference in the methodology, but some critical details about the CALIOP products used are missing. For example, which data version number, what quality screening was used, and what are the limitations of using the CALIOP retrievals as a reference? These details along with a summary of how the CALIOP extinction retrievals are calculated should be added as a subsection of the Section 2 (Data and methodology). Likewise, a summary of the AERONET data should be added to that section, including details on version number, which ground stations were evaluated, and quality screening information.

The analysis strategy is sound. The manuscript provides an objective summary of the intercomparisons and the conclusions are supported by the data presented. This manuscript is suitable for Atmospheric Measurement Techniques. Between the comments above and the specific comments below, I recommend minor revisions are needed prior to publication.

Thank you for your feedback. We agree that adding details will strengthen our paper. First, for the conversion of aerosol layer height definitions, we used a look-up table approach. As shown in Figure 2, we highlight the differences in aerosol layer height definitions based on height, which lead to the development of our conversion look-up table. Throughout this paper, we conducted two conversions: first, we aligned all passive sensor aerosol layer height products with the CALIOP AOCH definition for comparison, as illustrated in Figure 1b. Second, we converted GEMS AEH to the EPIC and TROPOMI AOCH definition, based on Figure 1c. These details are added at the end of Section 2.2 as follows:

*In our further comparison of ALH, we count for these inherent differences by converting one definition to another to ensure consistency. Varying AOCH from 0-10 km, we created a look-up table of AEH and, EPIC/TROPOMI AOCH, and CALIOP AOCH corresponding to the same aerosol extinction profile according to their different definitions. Throughout this paper, we conducted two conversions to ensure consistency: first, we converted all passive sensor ALH products with the CALIOP AOCH definition for comparison with CALIOP data (Fig. 1b). Second, we converted GEMS AEH to the EPIC and TROPOMI AOCH definition for comparisons among passive remote sensing products (Fig 1c).*

Additionally, we changed the title of Section 2 from 'Satellite data' to 'Remote sensing data' and added 2.1.4 CALIOP / CALIPSO and 2.1.5 AERONET as follows:

*2.1 Remote sensing data*

*2.1.4 CALIOP / CALIPSO*

*CALIOP is a lidar system on the CALIPSO platform that provides attenuated backscatter vertical profiles of aerosols and clouds in the atmosphere using a two-wavelength laser operating at 532 nm with linear polarization and at 1064 nm (Winker et al., 2009). While the global coverage of CALIOP is*

*less than 0.2%, it provides high vertical resolution for retrieving aerosol extinction profiles (Winker et al., 2013). In this paper, we used CALIOP 5 km Level 2 aerosol extinction profile product at 532 nm to derive optical depth weighted heights. Specifically, Level 2 Aerosol Profile, Version 4-21 data product for the year 2021 (CAL_LID_L2_05kmAPro-Standard-V4-21) is used. For the years 2022 to 2023, level 2 aerosol profile, version 4-51 (CAL_LID_L2_05kmAPro-Standard-V4-51) is used due to the data availability. To validate aerosol height retrievals from passive remote sensing with CALIOP observation, the optical depth weighted heights derived from CALIOP 5 km level 2 aerosol extinction profile product at 532 nm following previous studies are used (Chen et al., 2021b; Lu et al., 2023; Lu et al., 2021; Xu et al., 2019).*

*2.1.5 AERONET*

*AERONET is a ground-based remote sensing network to designed to measure and characterize aerosol optical properties through direct sun measurements taken with sun-sky scanning spectral radiometers (Holben et al., 1998). AERONET serves as a critical tool for validating satellite-retrieved aerosol optical properties including AOD. In this study, we used AOD data at 675 nm and 440 nm from AERONET Version 3 Level 1.5 to assess the accuracy of satellite AOD retrievals. AERONET sites located within our study domain of East Asia and Southeast Asia, as illustrated in Figure 2, were selected for this analysis. Additional information of these sites can be found in S1.*

Line 57. Comma not needed "present, from"

Comma deleted.

Line 79. The grammar is a little off here. I think it should be "Not only are these techniques based on different physical theories,"

Modified.

Section 2. Background information on the CALIOP data used and AERONET data used should be added to this section. Version numbers, specific products, ground stations (for AERONET), quality screening, etc…)

We added 2.1.4 CALIOP / CALIPSO and 2.1.5 AERONET in the Section 2.

Line 234. Is there a term for σH? In other words, is there a name for this value? This is not critical but might make the description clearer.

$\sigma_H$ *is the half width parameter defined as:* $\sigma_H = ln(3 + \sqrt{8})/\eta$

Lines 235-236. "EPIC and TROPOMI defined their retrieved ALH as H in Eq. (1) and called AOCH" Should this sentence say, "…and it is called AOCH"? I think this might make the sentence clearer (or another suitable modification).

Modified to:

*EPIC and TROPOMI defined their retrieved ALH as H in Equation (1) and referred to it as AOCH.*

Line 243. This is called the "CALIOP 5 km level 2 aerosol profile product". It contains profiles of aerosol extinction. This and the references in the sentence should be moved to a Section 2 subsection describing the CALIOP data used.

We moved this information to 2.1.4 CALIOP / CALIPSO and 2.1.5 AERONET.

Line 253. It would be more precise to say "reaches approximately 4 km and above" rather than "4 km and beyond".

Modified.

Lines 258-259. "In our further comparison of ALH,…by converting one definition to another" More details should be added to explain how this is accomplished. My understanding is that the passive sensors report AHL altitudes (with different definitions). How are these altitudes converted?

*Answered above in general comments.*

Figures 4, 5, 7 captions. Add text to the captions explaining what the gray points indicate.

*Grey points indicate points where the data density is less than 0.01.*

Lines 391-392. "the ALH values of all passive sensors are converted to AOCH following the CALIOP AOCH definition". Add the details of how they are converted (here or another suitable location).

*Answered above in general comments.*

Lines 444-445. " GEMS AEH is converted to align with the EPIC and TROPOMI AOCH definition". How is this done?

*Answered above in general comments.*

Line 457. "This is a combination of inaccurate cloud detection and inherent sensitivity in the retrieval process." Because the previous sentence involves three different instruments, be specific on which instruments are influenced by inaccurate cloud detection and sensitivity.

*This is a combination of inaccurate cloud detection and inherent sensitivity in the retrieval process of TROPOMI and EPIC.*

Figure 7 caption. It is not clear which panels are dust and which are smoke. An option could be to say "dust (top row) and smoke (bottom row)"

All panels of Figure 7 are dust and smoke cases combined (all cases). The top row is for GEMS and TROPOMI comparison and the bottom row is for GEMS and EPIC comparison. We changed the caption for clarity.

*Intercomparison of AOCH values from GEMS, TROPOMI, and EPIC for all cases (dust and smoke combined) as a function of UVAI. The density scatter plots show the AOCH comparison between GEMS and TROPOMI (a – c), and between GEMS and EPIC (d – f). (b) and (e) represent GEMS data for UVAI < 3, while (e) and (f) represent data for UVAI ≥ 3. GEMS AEH values have been converted to align with the AOCH definitions used by EPIC and TROPOMI.*

Line 476. "GEMS and EPIC products for the day are adjusted to relative local solar time (LST) accordingly." Please rephrase to clarify: the wording makes it sound like the ALH data is adjusted, but I think it is the time of observation that is converted to local time. Is that right? It is not clear. If the actual ALH data is adjusted then an explanation should be added about how this adjustment is made.

How you understood is correct. We rephrased to clarify the time of observation is converted to local time:

*Therefore, we define the relative local solar noon time for a given day as the moment when the solar zenith angle at a particular location reaches its minimum value. Using this relative local solar noon as a reference, we adjust the observation times of GEMS and EPIC products with relative local solar time (LST).*

Figure 9 and 12. The size of the star is obscuring the ability to see the data underneath. Recommend reducing its size. A description of the star should probably be added to the captions too instead of only the text (e.g., "black star indicates center of study domain")

We changed the colormap of Figure 9 to the same as that of Figure 12 (according to the comment below) as well as the start color from black to red.

*The red star indicates the dust plume area near Beijing.*

Figures 9 and 12. Recommend using the same colormap for these images since they are the same quantity. The figure 12 colormap is better given the sequential nature of the data. This an optional recommendation (though I am not sure if the AMT style guide has any rules about this).

We updated the colormap of Figure 9 to match that of Figure 12, as it is more accessible for individuals with color vision deficiencies and adheres to AMT's stylistic guidelines.

Figure 13 caption. Should read, "Same as figure 10" instead of "Same as figure 11"

Yes, thank you for the correction.

Lines 639-640. "…further compounded by inaccurate surface reflectance in TROPOMI and EPIC." This was conjectured in Section 3.1, but not proven. ("The positive y-intercept observed for both EPIC and TROPOMI suggests that the surface reflectance employed in the dust aerosol model may be underestimated…"). The positive y-intercept is consistent with an underestimate of the surface reflection, but other causes where not discussed and ruled out. The statement in lines 639-640 should be modified to say that this is a possible cause that is consistent with the observations in this study.

*The inaccuracies in the GEMS dust aerosol model contribute to the significant differences in GEMS AOD compared to TROPOMI and EPIC. Additionally, the differences are compounded by other causes including potential inaccuracies in surface reflectance in TROPOMI and EPIC.*

Lines 656-661. This paragraph summarizes the intercomparisons of ALH between CALIOP and the three passive instruments. It contrasts the correlations and biases between EPIC, TROPOMI, and GEMS with respect to the co-located CALIOP observation based on the statistics reported in Figs. 10a and 13a. I do not think the passive sensors are being equitably compared because all three passive sensors do not always report ALH values in every profile where CALIOP reports an ALH. For example, in Fig. 10, only GEMS reports ALH south of 35°N. However, those profiles contribute to the statistics for GEM-CALIOP comparisons in Fig. 10a. EPIC and TROPOMI had no ALH reported in these profiles so their comparison statics do not consider this part of the scene. To fairly compare the three passive instruments with respect to CALIOP would be to only compare statistics based on profiles where all three instruments and CALIOP reported ALH. Figures 10a and 13a could probably stay as they are (scatter plot based on the whole scene), however in this section of the conclusions where general statements are being made about the relative accuracy of each instrument compared to another, they should each have ALH values to report.

We agree that it will be helpful to compare ALH values only where all retrieval products (EPIC, TROPOMI, and GEMS) have valid data. However, it is important to highlight the distinct retrieval characteristics of each instrument. While TROPOMI and EPIC AOCH primarily provide retrievals for absorbing aerosols, GEMS retrieves both absorbing and non-absorbing aerosols. This difference reflects the inherent variations in the algorithms of each product and is a critical aspect of the intercomparison.

To address the concern, we have included a supplementary scatter plot where all three instruments and CALIOP report valid ALH data. In this subset, we observe significantly high correlations (R > 0.9) between the passive sensors and CALIOP ALH. Specifically, GEMS demonstrates the lowest RMSE (0.38 km), while EPIC and TROPOMI show larger RMSE values of 1.54 km and 1.11 km, respectively, with a tendency to overestimate ALH.

The paragraph is summarizing for individual dust and smoke case studies. Therefore, this additional comparison is now reflected in the 3.3 ALH validation with CALIOP, conclusion, and the supplementary (Fig. S6).

[Figure]

*When valid data are available from all retrievals, the passive sensors show a notably high correlation with CALIOP AOCH (R > 0.9), with GEMS having the lowest RMSE (0.38 km), while EPIC and TROPOMI exhibit higher RMSE values of 1.54 km and 1.11 km, respectively (Fig. S6)*

Appendix A. List of acronyms should be alphabetized

We alphabetized the list of acronyms.

**Report #2 Anonymous referee #4**

Dear Authors,

Thank you for this well-written and extremely detailed manuscript describing your comparative analysis of aerosol layer height retrievals from multiple satellite sensors. The potential to characterize the vertical position of aerosol plumes from space using passive sensors is a very exciting one with important applications for air quality monitoring and other areas.

I think your scientific analysis is fundamentally sound. I have some recommendations for clarifying your account, and a few places where rewording is needed to avoid misunderstanding of your results. I believe with these minor edits, this manuscript can be ready for publication.

I have a few more substantial comments, with minor grammar and typographic edits at the bottom.

Line 478: "descends… indicating the deposition process" this is a statement without evidence. Deposition is a process at the surface that removes aerosols from the atmosphere; the process you are thinking of is "gravitational settling" sometimes called "sedimentation." For this case, you could say that this change in retrieved ALH reflects either gravitational settling or atmospheric subsidence, but the actual cause is almost certainly subsidence.

Thank you for your insight and suggestion. We believe that using the term "subsidence" would be the most accurate for our case.

*GEMS measurements show that the dust plume is located at 4-5 km on March 27, descends to ~3 km on March 28, and remains at that altitude until March 29, consistent with EPIC and TROPOMI measurements. These daily changes in ALH reflects the atmospheric subsidence of dust aerosols during transport.*

Figure 9 (and Figure 11): You have hourly GEMS retrievals, showing a descending plume, and a series of looks from EPIC and TROPOMI, which do not indicate any trend in aerosol layer height. You summarize this by stating that "Likely, multiple GEMS observations reveal clear hourly variations, whereas discerning diurnal changes is challenging with TROPOMI and EPIC." You need to make a clearer statement here—is the difference reflective of different retrieval sampling (TROPOMI and EPIC obviously have large gaps in the observations compared with GEMS), or is it due to differences in the retrieved quantities?

The original statement "Likely, multiple GEMS observations reveal clear hourly variations, whereas discerning diurnal changes is challenging with TROPOMI and EPIC." is written to explain one dust plume case on a March 28, 2021. In this particular case, TROPOMI and EPIC provide fewer observations compared to hourly GEMS observations. We have revised the statement as follows:

*For this dust case, hourly GEMS observations reveal clear hourly variations, while characterizing diurnal changes from TROPOMI and EPIC is challenging due to their limited number of observations compared to GEMS.*

Introduction Lines 60-70: I am not sure what this summary of Choi et al. 2021 is doing here. The finding of Choi et al. was that only with polarization-sensitive measurements did the DOFS increase to permit largely independent retrieval of aerosol layer height and thickness—still not enough information for actual multi-layer cases. As you describe, none of the sensors you are using have this polarization sensitivity, and so they retrieve only a single layer height parameter using an assumed layer thickness. I think this section could be both shorter and more clear.

We decided to remove this reference as it may not be directly relevant to our study.

Line 295-305: for comparisons where there is significant mean bias between the samples, the linear regression slope and intercept are not very robust; I recommend relying mostly on r2 and RMSE for this analysis.

We removed the analysis from slope and intercept and rewrote it as follows:

*For dust cases, both TROPOMI and EPIC AOD show a positive bias compared to AERONET AOD, with values of 0.23 and 0.33 for TROPOMI and EPIC, respectively. In contrast to the dust cases, TROPOMI and EPIC AOD exhibit a negligible bias and smaller RMSE for smoke cases. Although TROPOMI and EPIC AOD do not provide retrievals for values less than 0.2, many AERONET AOD data points exist with values under this threshold, particularly in dust cases. This suggests that the surface reflectance used in the dust aerosol model from EPIC and TROPOMI may be underestimated, resulting in an overestimation in AOD retrieval.*

Line 285-295: If I understand correctly, you are saying that AOD for smoke cases correlates equally well with AERONET at 688 and 443 nm, while for dust cases, the correlations are weaker, and the RMSE higher, at 688nm. This could be written more clearly.

Correct. We clarified the writing:

GEMS AOD at 443 nm exhibits a strong positive correlation with AERONET AOD at 440 nm, with correlation coefficients (R) of 0.9 for dust cases and 0.88 for smoke cases (Fig. 3a). At 680 nm, the correlation for smoke cases remains high at R = 0.84, indicating a similar level of agreement with the 443 nm measurements. However, for dust cases at 680 nm, the correlation decreases to R = 0.73, along with a 17 % increase in RMSE, indicating distinct retrieval accuracy of GEMS AOD at 443 and 680 nm for dust.

Abstract, line 30: I think you mean "overestimate ALH"?

It is meant to be AOD, but "km" was incorrectly written. This was corrected in newer version of the manuscript from the first revision.

Line 161 "eastern half" "western region"

Corrected.

Multiple places: I recommend describing EPIC sampling as "near-hourly" rather than "close-hourly"

We changed all "close-hourly" to "near-hourly".

Line 170: "The fact that the lower surface reflectance…motives us" => "Lower surface reflectance in the O2 B band vs the O2 A band suggests that O2 B band can be used to improve ALH retrievals using A band only."

Modified to:

*Lower surface reflectance in O2 B band compared to O2 A band over land (Xu et al., 2019) suggests that O2 B band can be used to improve the ALH retrievals using O2 A band only.*

Line 187: "this spectral range includes many trace gas absorption bands, and is also sensitive to aerosols and surface properties."

Corrected as follows:

*TROPOMI on board the Copernicus Sentinel-5 Precursor satellite was launched in October 2017 to measure solar radiation reflected by Earth from UV to shortwave infrared (SWIR) bands. This spectral range includes many trace gas absorption bands, and is also sensitive to aerosols and surface properties.*

Line 202: "Less valid retrievals" => "Fewer valid retrievals"

This part has been removed.

Line 320-321: "The inaccuracy of the GEMS dust aerosol model… has a notable impact"

Corrected grammar.